# Auditing Local Explanations is Hard

**Robi Bhattacharjee**
University of Tübingen and Tübingen AI Center
`robi.bhattacharjee@wsii.uni-tuebingen.de`

**Ulrike von Luxburg**
University of Tübingen and Tübingen AI Center
`ulrike.luxburg@uni-tuebingen.de`

## Abstract

In sensitive contexts, providers of machine learning algorithms are increasingly required to give explanations for their algorithms' decisions. However, explanation receivers might not trust the provider, who potentially could output misleading or manipulated explanations. In this work, we investigate an auditing framework in which a third-party auditor or a collective of users attempts to sanity-check explanations: they can query model decisions and the corresponding local explanations, pool all the information received, and then check for basic consistency properties. We prove upper and lower bounds on the amount of queries that are needed for an auditor to succeed within this framework. Our results show that successful auditing requires a potentially exorbitant number of queries – particularly in high dimensional cases. Our analysis also reveals that a key property is the "locality" of the provided explanations — a quantity that so far has not been paid much attention to in the explainability literature. Looking forward, our results suggest that for complex high-dimensional settings, merely providing a pointwise prediction and explanation could be insufficient, as there is no way for the users to verify that the provided explanations are not completely made-up.

## 1 Introduction

Machine learning models are increasingly used to support decision making in sensitive contexts such as credit lending, hiring decisions, admittance to social benefits, crime prevention, and so on. In all these cases, it would be highly desirable for the customers/applicants/suspects to be able to judge whether the model's predictions or decisions are "trustworthy". New AI regulation such as the European Union's AI Act can even legally require this. One approach that is often held up as a potential way to achieve transparency and trust is to provide *local explanations*, where every prediction/decision comes with a human-understandable explanation for this particular outcome (e.g., LIME (Ribeiro et al., 2016), SHAP (Lundberg and Lee, 2017), or Anchors (Ribeiro et al., 2018)).

However, in many real-world scenarios, the explanation receivers may not necessarily trust the explanation providers (Bordt et al., 2022). Imagine a company that uses machine learning tools to assist in screening job applications. Because the company is well-advised to demonstrate fair and equitable hiring, it is plausible that it might bias its explanations towards depicting these properties. And this is easy to achieve: the company is under full control of the machine learning model and the setup of the explanation algorithm, and prior literature (Ghorbani et al., 2019; Dombrowski et al., 2019) has shown that current explainability tools can be manipulated to output desirable explanations.

This motivates the question: what restrictions or procedures could be applied to prevent such explanation cheating, and more specifically, what are ways to verify that the provided explanations

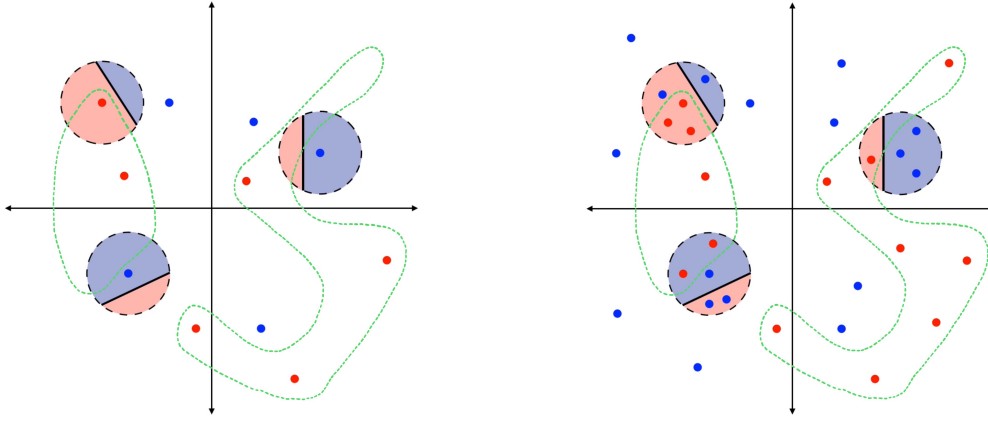

(a) Insufficient data for auditing     (b) Sufficient data for auditing

Figure 1: **Local explanations** (see Section 2.2 for notation): In both panels, a set of training points $x$ and their classifications $f(x)$ (red/blue, decision boundary in green) are shown. For three training points (one centered at each ball), a local linear explanation $(g_x, R_x)$ is illustrated where $g_x$ is a local linear classifier (black decision boundary) and $R_x$ is a local ball centered at $x$. *Panel (a)* depicts a regime where there is *insufficient* data for verifying how accurate the local explanations approximate the classifier $f$ – none of the provided regions contain enough points to assess the accuracy of the linear explanations. *Panel (b)* depicts a regime with more training points allowing us to validate the accuracy of the linear explanations based on how closely they align with the points in their corresponding regions.

are actually trustworthy? One approach is to require that the explanation providers completely publicize their models, thus allowing users or third-party regulators to verify that the provided explanations are faithful to the actual model being used. However, such a requirement would likely face stiff resistance in settings where machine learning models are valuable intellectual property.

In this work, we investigate an alternative approach, where a third-party regulator or a collective of users attempt to verify the trustworthiness of local explanations, simply based on the predictions and explanations over a set of examples. The main idea is that by comparing the local explanations with the actual predictions across enough data one could, in principle, give an assessment on whether the provided explanations actually adhere to the explained model. The goal of our work is to precisely understand when this is possible.

### 1.1   Our contributions: data requirements for auditing.

We begin by providing a general definition for local explainability that encompasses many popular explainability methods such as Anchors (Ribeiro et al., 2018), Smooth-grad (Smilkov et al., 2017), and LIME (Ribeiro et al., 2016). We define a *local explanation* for a classifier $f$ at a point $x$ as a pair $(R_x, g_x)$, where $R_x$ is a local region surrounding $x$, and $g_x$ is a simple local classifier designed to approximate $f$ over $R_x$. For example, on continuous data, Anchors always output $(R_x, g_x)$ where $R_x$ is a hyper-rectangle around $x$ and $g_x$ is a constant classifier; gradient-based explanations such as Smooth-grad or LIME implicitly approximate the decision function $f$ by a linear function in a local region around $x$.

Obviously, any human-accessible explanation that is being derived from such a local approximation can only be trustworthy if the local function $g_x$ indeed approximates the underlying function $f$ on the local region $R_x$. Hence, a *necessary condition* for a local explanation to be trustworthy is that the function $g_x$ is close to $f$ on the region $R_x$, and this should be the case for most data points $x$ sampled from the underlying distribution.

To measure how closely a set of local explanations adheres to the original classifier $f$, we propose an explainability loss function $L_\gamma(E, f)$, which quantifies the frequency with which $f$ differs by more

than $\gamma$ from the local classifier $g_x$ over the local region $R_x$ (see Sec. 2.2 for precise definitions). We then introduce a formalism for *auditing local explanations* where an auditor attempts to estimate the explainability loss $L_\gamma(E, f)$. In our formalism, the auditor does so with access to the following objects:

1. A set of data points $X = \{x_1, \ldots, x_n\}$ drawn i.i.d from the underlying data distribution.
2. The outputs of a classifier on these points, $f(X) = \{f(x_1), \ldots, f(x_n)\}$.
3. The provided local explanations for these points $E(f, X) = \{E(f, x_1), \ldots, E(f, x_n)\}$

Observe that in our formalism, the auditor has only restricted access to the machine learning model and the explanations: they can only interact with them through their evaluations at specific data-points. We have chosen this scenario because we believe it to be the most realistic one in many practical situations, where explanation providers try to disclose as little information on their underlying machine learning framework as possible.

In our main result, Theorem 4.1, we provide a lower bound for the amount of data needed for an auditor to accurately estimate $L_\gamma(E, f)$. A key quantity in our analysis is the *locality* of the provided explanations. We show that the smaller the provided local regions $R_x$ are, the more difficult it becomes to audit the explainer. Intuitively, this holds because estimating the explainability loss relies on observing multiple points within these regions, as illustrated in Panel (b) of Figure 1. By contrast, if this fails to hold (Panel (a)), then there is no way to validate how accurate the local explanations are. We also complement our lower bound with an upper bound (Theorem 4.2) that demonstrates that reasonably large local regions enable auditing within our framework.

Our results imply that the main obstacle to auditing local explanations in this framework is the locality of the provided explanations. As it turns out, this quantity is often *prohibitively small in practice*, making auditing *practically impossible*. In particular, for high-dimensional applications, the local regions $R_x$ given by the explainer are often exponentially small in the data-dimension. Thus the explanations cannot be verified in cases where there does not exist any prior trust between the explanation provider and the explanation receivers.

We stress that estimating the local loss $L_\gamma(E, f)$ serves as a first *baseline* on the path towards establishing trustworthy explanations. It is very well possible that an explanation provider achieves a small local loss (meaning that the local classifiers closely match the global classifier $f$) but nevertheless provides explanations that are misleading in some other targeted manner. Thus, we view successful auditing in this setting as a *necessary but not sufficient* condition for trusting an explanation provider.

Our results might have far-reaching practical consequences. In cases where explanations are considered important or might even be required by law, for example by the AI Act, it is a necessary requirement that explanations can be verified or audited (otherwise, they would be completely useless). Our results suggest that in the typical high dimensional setting of modern machine learning, auditing pointwise explanations is impossible if the auditor only has access to pointwise decisions and corresponding explanations. In particular, collectives of users, for example coordinated by non-governmental organizations (NGOs), are never in the position to audit explanations. The only way forward in auditing explanations would be to appoint a third-party auditor who has more power and *more access to the machine learning model*, be it access to the full specification of the model function and its parameters, or even to the training data. Such access could potentially break the fundamental issues posed by small local explainability regions in our restricted framework, and could potentially enable the third party auditor to act as a moderator to establish trust between explanation receivers and explanation providers.

## 1.2 Related Work

Prior work (Yadav et al., 2022; Bhatt et al., 2020; Oala et al., 2020; Poland, 2022) on auditing machine learning models is often focused on applying explainability methods to audit the models, rather than the explanations themselves. However, there has also been recent work (Leavitt and Morcos, 2020; Zhou et al., 2021) arguing for more rigorous ways to evaluate the performance of various explanation methods. There are numerous approaches for doing so: including performance based on human-evaluation (Jesus et al., 2021; Poursabzi-Sangdeh et al., 2021), and robustness (Alvarez-Melis and Jaakkola, 2018).

There has also been a body of work that evaluates explanations based on the general notion of faithfulness between explanations and the explained predictor. Many approaches (Wolf et al., 2019; Poppi et al., 2021; Tomsett et al., 2020) examine neural-network specific measures, and typically rely on access to the neural network that would not be present in our setting. Others are often specialized to a specific explainability tool – with LIME (Visani et al., 2022; Botari et al., 2020) and Shap (Huang and Marques-Silva, 2023) being especially popular choices.

By contrast, our work considers a general form of local explanation, and studies the problem of auditing such explanations in a restricted access setting, where the auditor only interacts with explanations through queries. To our knowledge, the only previous work in a similar setting is (Dasgupta et al., 2022), in which local explanations are similarly audited based on collecting them on a set of data sampled from a data distribution. They consider a quantity called the *local sufficiency*, which directly corresponds to our notion of local loss (Definition 2.3). However, their work is restricted to a *discrete* setting where local fidelity is evaluated based on instances that receive *identical* explanations. In particular, they attempt to verify that points receiving identical explanations also receive identical predictions. By contrast, our work lies within a *continuous* setting, where a local explanation is said to be faithful if it matches the underlying model over a *local region*.

A central quantity to our analysis is the *locality* of an explanation, which is a measure of how large the local regions are. Prior work has rarely measured or considered this quantity, with a notable exception being Anchors method (Ribeiro et al., 2018) which utilizes it to assist in optimizing their constructed explanations. However, that work did not explore this quantity beyond treating it as a fixed parameter.

Finally, we note that other recent work, such as (Bassan and Katz, 2023), provides avenues for providing explanations with *certifiable correctness*, meaning that they provide proof that their accurate reflect the underlying model. We view our work as complementary to such methods as our work demonstrates the *necessity* of such ideas by demonstrating difficulties with using *generic local explanation methods*.

## 2 Local Explanations

### 2.1 Preliminaries

In this work, we restrict our focus to *binary classification* – we let $\mu$ denote a data distribution over $\mathbb{R}^d$, and $f : \mathbb{R}^d \to \{\pm 1\}$ be a so-called black-box binary classifier that needs to be explained. We note that lower bounds shown for binary classification directly imply lower bounds in more complex settings such as multi-class classification or regression.

For any measurable set, $M \subseteq \mathbb{R}^d$, we let $\mu(M)$ denote the probability mass $\mu$ assigns $M$. We will also let $supp(\mu)$ denote the *support* of $\mu$, which is the set of all points $x$ such that $\mu(\{x' : ||x - x'|| \le r\}) > 0$ for all $r > 0$.

We define a **hyper-rectangle** in $\mathbb{R}^d$ as a product of intervals, $(a_1, b_1] \times \cdots \times (a_d, b_d]$, and let $\mathcal{H}_d$ denote the set of all hyper-rectangles in $\mathbb{R}^d$. We let $\mathcal{B}_d$ denote the set of all $L_2$-balls in $\mathbb{R}^d$, with the ball of radius $r$ centered at point $x$ being defined as $B(x, r) = \{x' : ||x - x'|| \le r\}$.

We will utilize the following two simple hypothesis classes: $\mathcal{C}_d$, which is the set of the two constant classifiers over $\mathbb{R}^d$, and $\mathcal{L}_d$, which is the set of all linear classifiers over $\mathbb{R}^d$. These classes serve as important examples of *simple and interpretable classifiers* for constructing local explanations.

### 2.2 Defining local explanations and explainers

One of the most basic and fundamental concepts in Explainable Machine Learning is the notion of a *local explanation*, which, broadly speaking, is an attempt to explain a complex function's behavior at a specific point. In this section, we describe a general form that such explanations can take, and subsequently demonstrate that two widely used explainability methods, LIME and Anchors, adhere to it.

We begin by defining a *local explanation* for a classifier at a given point.

**Definition 2.1.** For $x \in \mathbb{R}^d$, and $f : \mathbb{R}^d \to \{\pm 1\}$, a **local explanation** for $f$ at $x$ is a pair $(R_x, g_x)$ where $R_x \subseteq \mathbb{R}^d$ is a region containing $x$, and $g_x : R_x \to \{\pm 1\}$ is a classifier.

Here, $g_x$ is typically a simple function intended to approximate the behavior of a complex function, $f$, over the region $R_x$. The idea is that the local nature of $R_x$ simplifies the behavior of $f$ enough to provide intuitive explanations of the classifier's local behavior.

Next, we define a *local explainer* as a map that outputs local explanations.

**Definition 2.2.** $E$ is a **local explainer** if for any $f : \mathbb{R}^d \to \{\pm 1\}$ and any $x \in \mathbb{R}^d$, $E(f, x)$ is a local explanation for $f$ at $x$. We denote this as $E(f, x) = (R_x, g_x)$.

We categorize local explainers based on the types of explanations they output – if $\mathcal{R}$ denotes a set of regions in $\mathbb{R}^d$, and $\mathcal{G}$ denotes a class of classifiers, $\mathbb{R}^d \to \{\pm 1\}$, then we say $E \in \mathcal{E}(\mathcal{R}, \mathcal{G})$ if for all $f, x$, $E(f, x)$ outputs $(R_x, g_x)$ with $R_x \in \mathcal{R}$ and $g_x \in \mathcal{G}$.

Local explainers are typically constructed for a given classifier $f$ over a given data distribution $\mu$. In practice, different algorithms employ varying amounts of access to both $f$ and $\mu$ – for example, SHAP crucially relies on data sampled from $\mu$ whereas gradient based methods often rely on knowing the actual parameters of the model, $f$. To address all of these situations, our work takes a black-box approach in which we make no assumptions about how a local explainer is constructed from $f$ and $\mu$. Instead we focus on understanding how to evaluate how effective a given explainer is with respect to a classifier $f$ and a data distribution $\mu$.

## 2.3 Examples of Explainers

We now briefly discuss how various explainability tools in practice fit into our framework of local explanations.

**Anchors:** The main idea of Anchors (Ribeiro et al., 2018) is to construct a region the input point in which the desired classifier to explain remains (mostly) constant. Over continuous data, it outputs a local explainer, $E$, such that $E(x) = (R_x, g_x)$, where $g_x$ is a constant classifier with $g_x(x') = f(x)$ for all $x' \in \mathbb{R}^d$, and $R_x$ is a hyper-rectangle containing $x$. It follows say that the Anchors method outputs an explainer in the class, $\mathcal{E}(\mathcal{H}_d, \mathcal{C}_d)$.

**Gradient-Based Explanations:** Many popular explainability tools (Smilkov et al., 2017; Agarwal et al., 2021; Ancona et al., 2018) explain a model's local behavior by using its gradient. By definition, gradients have a natural interpretation as a locally linear model. Because of this, we argue that gradient-based explanations are implicitly giving local explanations of the form $(R_x, g_x)$, where $R_x = B(x, r)$ is a small $L_2$ ball centered at $x$, and $g_x$ is a linear classifier with coefficients based on the gradient. Therefore, while the radius $r$ and the gradient $g_x$ being used will vary across explanation methods, the output can be nevertheless interpreted as an explainer in $\mathcal{E}(\mathcal{B}_d, \mathcal{L}_d)$, where $\mathcal{B}_d$ denotes the set of all $L_2$-balls in $\mathbb{R}^d$, and $\mathcal{L}_d$ denotes the set of all linear classifiers over $\mathbb{R}^d$.

**LIME:** At a high level, LIME (Ribeiro et al., 2016) also attempts to give local linear approximations to a complex model. However, unlike gradient-based methods, LIME includes an additional feature-wise discretization step where points nearby the input point, $x$, are mapped into a binary representation in $\{0, 1\}^d$ based on how similar a point is to $x$. As a consequence, LIME can be construed as outputting local explanations of a similar form to those outputted by gradient-based methods.

Finally, as an important limitation of our work, although many well-known local explanations fall within our definitions, this does not hold in all cases. Notably, Shapley-value (Lundberg and Lee, 2017) based techniques do not conform to the format given in Definition 2.1, as it is neither clear how to construct local regions that they correspond to, nor the precise local classifier being used.

## 2.4 A measure of how accurate an explainer is

We now formalize what it means for a local classifier, $g_x$, to "approximate" the behavior of $f$ in $R_x$.

**Definition 2.3.** For explainer $E$ and point $x$, we let the **local loss**, $L(E, f, x)$ be defined as the fraction of examples drawn from the region $R_x$ such that $g_x$ and $f$ have different outputs. More precisely, we set

$$L(E, f, x) = \Pr_{x' \sim \mu} [g_x(x') \neq f(x) | x' \in R_x].$$

$\mu$ is implicitly used to evaluate $E$, and is omitted from the notation for brevity. We emphasize that this definition is specific to *classification*, which is the setting of this work. A similar kind of loss can be constructed for regression tasks based on the mean-squared difference between $g_x$ and $f$.

We contend that maintaining a low local loss across most data points is *essential* for any reasonable local explainer. Otherwise, the explanations provided by the tool can be made to support any sort of explanation as they no longer have any adherence to the original function $f$.

To measure the overall performance of an explainer over an entire data distribution, it becomes necessary to aggregate $L(E, f, x)$ over all $x \sim \mu$. One plausible way to accomplish this would be to average $L(E, f, x)$ over the entire distribution. However, this would leave us unable to distinguish between cases where $E$ gives extremely poor explanations at a small fraction of points as opposed to giving mediocre explanations over a much larger fraction. To remedy this, we opt for a more precise approach in which a user first chooses a **local error threshold**, $0 < \gamma < 1$, such that local explanations that incur an explainabiliy loss under $\gamma$ are considered acceptable. They then measure the global loss for $E$ by determining the fraction of examples, $x$, drawn from $\mu$ that incur explainability loss above $\gamma$.

**Definition 2.4.** Let $\gamma > 0$ be a user-specified local error threshold. For local explainer $E$, we define the **explainability loss** $L_\gamma(E, f)$ as the fraction of examples drawn from $\mu$ that incur a local loss larger than $\gamma$. That is,

$$L_\gamma(E, f) = \Pr_{x \sim \mu}[L(E, f, x) \geq \gamma].$$

We posit that the quantity $L_\gamma(E, f)$ serves as an overall measure of how faithfully explainer $E$ adheres to classifier $f$, with lower values of $L_\gamma(E, f)$ corresponding to greater degrees of faithfulness.

## 2.5   A measure of how large local regions are

The outputted local region $R_x$ plays a crucial role in defining the local loss. On one extreme, setting $R_x$ to consist of a single point, $\{x\}$, can lead to a perfect loss of 0, as the explainer only needs to output a constant classifier that matches $f$ at $x$. But these explanations would be obviously worthless as they provide no insight into $f$ beyond its output $f(x)$. On the other extreme, setting $R_x = \mathbb{R}^d$ would require the explainer to essentially replace $f$ in its entirety with $g_x$, which would defeat the purpose of explaining $f$ (as we could simply use $g_x$ instead). Motivated by this observation, we define the *local mass* of an explainer at a point $x$ as follows:

**Definition 2.5.** The **local mass** of explainer $E$ with respect to point $x$ and function $f$, denoted $\Lambda(E, f, x)$, is the probability mass of the local region outputted at $x$. That is, if $E(f, x) = (R_x, g_x)$, then

$$\Lambda(E, f, x) = \Pr_{x' \sim \mu}[x' \in R_x].$$

Based on our discussion above, it is unclear what an ideal local mass is. Thus, we treat this quantity as a property of local explanations rather than a metric for evaluating their validity. As we will later see, this property is quite useful for characterizing how difficult it is to estimate the explainability loss of an explainer. We also give a global characterization of the local mass called *locality*.

**Definition 2.6.** The **locality** of explainer $E$ with respect to function $f$, denoted $\Lambda(E, f)$, is the minimum local mass it incurs. That is, $\Lambda(E, f) = \inf_{x \in supp(\mu)} \Lambda(E, f, x)$.

## 3   The Auditing Framework

Recall that our goal is to determine how explanation receivers can verify provided explanations in situations where there *isn't* mutual trust. To this end, we provide a framework for *auditing local explanations*, where an auditor attempts to perform this verification with as little access to the underlying model and explanations as possible. Our framework proceeds in with the following steps.

1. The auditor fixes a local error threshold $\gamma$.
2. A set of points $X = \{x_1, \ldots, x_n\}$ are sampled i.i.d from data distribution $\mu$.
3. A black-box classifier $f$ is applied to these points. We denote these values with $f(X) = \{f(x_1), \ldots, f(x_n)\}$.

4. A local explainer $E$ outputs explanations for $f$ at each point. We denote these explanations with $E(f, X) = \{E(f, x_1), \ldots, E(f, x_n)\}$.

5. The Auditor outputs an estimate $A(X, f(X), E(f, X))$ for the explainability loss.

Observe that the auditor can only have access to the the model $f$ and its corresponding explanations *through* the set of sampled points. Its only inputs are $X$, $f(X)$, and $E(f, X)$. In the context of the job application example discussed in Section 1, this would amount to auditing a company based on the decisions and explanations they provided over a set of applicants.

In this framework, we can define the sample complexity of an auditor as the amount of data it needs to guarantee an accurate estimate for $L_\gamma(E, f)$. More precisely, fix a data distribution, $\mu$, a classifier, $f$, and an explainer $E$. Then we have the following:

**Definition 3.1.** For tolerance parameters, $\epsilon_1, \epsilon_2, \delta > 0$, and local error threshold, $\gamma > 0$, we say that an auditor, $A$, has **sample complexity** $N(\epsilon_1, \epsilon_2, \delta, \gamma)$ with respect to $\mu, E, f$, if for any $n \geq N(\epsilon_1, \epsilon_2, \delta, \gamma)$, with probability at least $1 - \delta$ over $X = \{x_1, \ldots, x_n\} \sim \mu^n$, $A$ outputs an accurate estimate of the explainability loss, $L_\gamma(E, f)$. That is,

$$L_{\gamma(1+\epsilon_1)}(E, f) - \epsilon_2 \leq A(X, f(X), E(f, X)) \leq L_{\gamma(1-\epsilon_1)}(E, f) + \epsilon_2.$$

Next, observe that our sample complexity is specific to the distribution, $\mu$, the classifier, $f$, and the explainer, $E$. We made this choice to understand the challenges that different choices of $\mu$, $f$, and $E$ pose to an auditor. As we will later see, we will bound the auditing sample complexity using the locality (Definition 2.5), which is a quantity that depends on $\mu$, $f$, and $E$.

# 4 How much data is needed to audit an explainer?

## 4.1 A lower bound on the sample complexity of auditing

We now give a lower bound on the amount of data needed to successfully audit an explainer. That is, we show that for any auditor $A$ and any data distribution $\mu$ we can find some explainer $E$ and some classifier $f$ such that $A$ is highly likely to give an inaccurate estimate of the explainability loss. To state our theorem we use the following notation and assumptions. Recall that $\mathcal{H}_d$ denotes the set of hyper-rectangles in $\mathbb{R}^d$, and that $\mathcal{C}_d$ denotes the set of the two constant binary classifiers over $\mathbb{R}^d$. Additionally, we will include a couple of mild technical assumptions about the data distribution $\mu$. We defer a detailed discussion of them to Appendices A.3 and A.1. We now state our lower bound.

**Theorem 4.1** (lower bound on the sample complexity of auditing). *Let $\epsilon_1, \epsilon_2 < \frac{1}{48}$ be tolerance parameters, and let $\gamma < \frac{1}{3}$ be any local error threshold. Let $\mu$ be any non-degenerate distribution, and $\lambda > 0$ be any desired level of locality. Then for any auditor $A$ there exists a classifier $f : \mathbb{R}^d \to \{\pm 1\}$ and an explainer $E \in \mathcal{E}(\mathcal{H}_d, \mathcal{C}_d)$ such that the following conditions hold.*

*1. $E$ has locality $\Lambda(E, f) = \lambda$.*

*2. There exist absolute constants $c_0, c_1 > 0$ such that if the auditor receives*

$$n \leq \frac{c_0}{\max(\epsilon_1, \epsilon_2)\lambda^{1-c_1 \max(\epsilon_1, \epsilon_2)}}$$

*many points, then with probability at least $\frac{1}{3}$ over $X = \{x_1, \ldots, x_n\} \sim \mu^n$, $A$ gives an inaccurate estimate of $L_\gamma(E, f)$. That is,*

$$A(X, f(X), E(f, X)) \notin [L_{\gamma(1+\epsilon_1)}(E, f) - \epsilon_2, L_{\gamma(1-\epsilon_1)}(E, f) + \epsilon_2].$$

In summary, Theorem 4.1 says that auditing an explainer requires an amount of data that is *inversely proportional* to its locality. Notably, this result does not require the data-distribution to be adversarially chosen, and furthermore applies when the explainer $E$ can be guaranteed to have a remarkably simple form being in $\mathcal{E}(\mathcal{H}_d, \mathcal{C}_d)$.

**Proof intuition of Theorem 4.1:** The main intuition behind Theorem 4.1 is that estimating the local explainability loss, $L(E, f, x)$, requires us to observe samples from the regions $R_x$. This would allow us to obtain an empirical estimate of $L(E, f, x)$ by simply evaluating the fraction of points

from $R_x$ that the local classifier, $g_x$, misclassifies. This implies that the locality $\lambda$ is a limiting factor as it controls how likely we are to observe data within a region $R_x$.

However, this idea enough isn't sufficient to obtain our lower bound. Although the quantity $\Omega\left(\frac{1}{\lambda^{1-O(\epsilon)}}\right)$ does indeed serve as a lower bound on the amount of data needed to guarantee seeing a large number of points within a region, $R_x$, it is unclear what a sufficient number of observations within $R_x$ is. Even if we don't have enough points in any single region, $R_x$, to accurately estimate $L(E, f, x)$, it is entirely plausible that aggregating loose estimates of $L(E, f, x)$ over a sufficient number of points $x$ might allow us to perform some type of estimation of $L_\gamma(E, f)$.

To circumvent this issue, the key technical challenge is constructing a distribution of functions $f$ and fixing $m = O\left(\frac{1}{\epsilon}\right)$ such that observing fewer than $m$ points from a given region, $R_x$, actually provides *zero information* about which function was chosen. We include a full proof in Appendix A.

### 4.2 An upper bound on the sample complexity of auditing.

We now show that if $\lambda$ is reasonably large, then auditing the explainability loss $L_\gamma(E, f)$ can be accomplished. As mentioned earlier, we stress that succeeding in our setting is *not* a sufficient condition for trusting an explainer – verifying that the local explanations $g_x$ match the overall function $f$ is just one property that a good explainer would be expected to have. Thus the purpose of our upper bound in this section is to complement our lower bound, and further support that the locality parameter $\lambda$ is the main factor controlling the sample complexity of an auditor.

Our auditing algorithm proceeds by splitting the data into two parts, $X_1$ and $X_2$. The main idea is to audit the explanations given for points in $X_1$ by utilizing the data from $X_2$. If we have enough data, then it is highly likely for us to see enough points in each local region to do this. We defer full details for this procedure to Appendix B.1. We now give the an upper bound on its sample complexity.

**Theorem 4.2** (**Upper Bound on Sample Complexity of Algorithm 1**). *There exists an auditor, A, for which the following holds. Let $\mu$ be a data distribution, $f$ be a classifier, and $E$ be an explainer. Suppose that $E$ has locality $\lambda$ with respect to $\mu$ and $f$. Let $\epsilon_1, \epsilon_2, \delta$ be tolerance parameters and let $\gamma > 0$ be a local error threshold. Then $A$ has sample complexity at most*

$$N(\epsilon_1, \epsilon_2, \delta, \gamma) = \tilde{O}\left(\frac{1}{\epsilon_2^2} + \frac{1}{\lambda\gamma^2\epsilon_1^2}\right).$$

This bound shows that the locality is sufficient for bounding the sample complexity for auditing local explanations. We defer a full proof to Appendix B. Observe that the dependency on $\lambda$ is $O(\frac{1}{\lambda})$ which matches the dependency in our lower bound provided that $\epsilon_1, \epsilon_2 \to 0$.

## 5 The locality of practical explainability methods can be extremely small

Theorems 4.1 and 4.2 demonstrate that the locality $\lambda$ characterizes the amount of data needed for an Auditor to guarantee an accurate estimate of the explainability loss $L_\lambda(E, f)$. It follows that if $\lambda$ is extremely small, then auditing could require a prohibitive amount of data. This leads to the following question: how small is $\lambda$ for practical explainability algorithms? To answer this, we will examine examine several commonly used algorithms that adhere to our framework.

We begin with **gradient-based methods**, which can be construed as providing an explainer in the class $\mathcal{E}(\mathcal{B}_d, \mathcal{L}_d)$, where $\mathcal{B}_d$ denotes the set of $L_2$ balls in $\mathbb{R}^d$, and $\mathcal{L}_d$ denotes the set of linear classifiers. To understand the impact of dimension on the locality of such explainers, we begin with a simple theoretical example.

Let $\mu$ be the data distribution over $\mathbb{R}^d$ that is a union of three concentric spheres. Specifically, $x \sim \mu$ is equally likely to be chosen at uniform from the sets $S_1 = \{x : ||x|| = 1 - \alpha\}$, $S_2 = \{x : ||x|| = 1\}$, and $S_3 = \{x : ||x|| = 1 + \beta\}$, where $\alpha, \beta$ are small $d$-dependent constants (Defined in Appendix C). Let $f : \mathbb{R}^d \to \{\pm 1\}$ be any classifier such that $f(x) = 1$ if $x \in S_1 \cup S_3$ and $f(x) = -1$ if $x \in S_2$. Observe that $\mu$ is a particularly simple data distribution over three spherical manifolds, and $f$ is a simple classifier that distinguishes its two parts. We illustrate this distribution in panel (a) of Figure 2.

Despite its simplicity, locally explaining $f$ with *linear* classifiers faces fundamental challenges. We illustrate this in Figure 2. Choosing a large local neighborhood, as done at point A, leads to issues

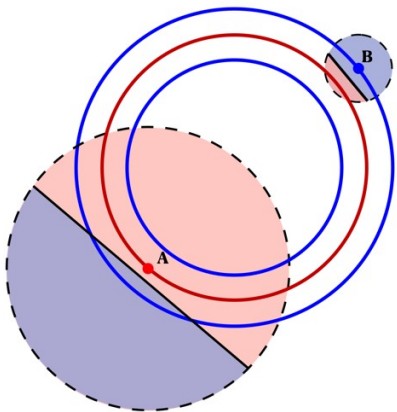

Figure 2: An illustration of Theorem 5.1, with the concentric blue and red circles depicting the data distribution $\mu$ classified by $f$, and with local explanations being depicted at points A and B. Explanations are forced to either have large local loss (point A) or a low local mass (point B).

posed by the curvature of the data distribution, meaning that it is impossible to create an accurate local linear classifier. On the other hand, choosing a neighborhood small enough for local linearity, as done in point B, leads to *local regions that are exponentially small with respect to the data dimension*. We formalize this in the following theorem.

**Theorem 5.1** (**A high dimensional example**). *Let $\mu, f$, be as described above, and let $E$ be any explainer in $\mathcal{E}(\mathcal{B}_d, \mathcal{L}_d)$. Let $x^*$ be any point chosen on the outer sphere, $S_3$. Then $E$ outputs an explanation at $x^*$ that either has a large local loss, or that has a small local mass. That is, either $L(E, f, x^*) \geq \frac{1}{6}$, or $\Lambda(E, f, x) \leq 3^{1-d}$.*

Theorem 5.1 demonstrates that if a locally linear explanation achieves even a remotely reasonable local loss, then it necessarily must have an extremely small local explanation. This suggests that, gradient based explanations will be exponentially local with respect to the data dimension, $d$.

We believe that this is also exhibited in practice particularly over *image data*, where explanations are often verified based on perceptual validity, rather than relevance to practical training points beyond the point being explained. For example, the explanations given by SmoothGrad (Smilkov et al., 2017) are visualized as pixel by pixel saliency maps. These maps often directly correspond to the image being explained, and are clearly *highly specific* to the it (see e.g. Figure 3 of (Smilkov et al., 2017)). As a result, we would hardly expect the implied linear classifier to have much success over almost any other natural image. This in turn suggest that the locality would be extremely small. We also remark that a similar argument can be made for **Lime**, which also tends to validate its explanations over images perceptually (for example, see Figure 4 of Ribeiro et al. (2016)).

Unlike the previous methods, **Anchors** (Ribeiro et al., 2018) explicitly seeks to maximize the local mass of its explanations. However, it abandons this approach for image classifiers, where it instead maximizes a modified form of locality based on super-imposing pixels from the desired image with other images. While this gives perceptually valid anchors, the types of other images that fall within the local region are completely unrealistic (as illustrated in Figure 3 of (Ribeiro et al., 2018)), and the true locality parameter is consequently extremely small. Thus, although Anchors can provide useful and *auditable* explanations in low-dimensional, tabular data setting, we believe that they too suffer from issues with locality for high-dimensional data. In particular, we note that it is possible to construct similar examples to Theorem 5.1 that are designed to force highly local Anchors-based explanations.

## 6   Conclusion

Our results in Section 4 demonstrate that the locality of a local explainer characterizes how much data is needed to audit it; smaller local regions lead to larger amounts of data. Meanwhile, our

discussion in Section 5 shows that typical local explanations are *extremely local* in high-dimensional space. It follows that in many cases, auditing solely based on point-wise decisions and explanations is impossible. Thus, any entity without model access, such as a collective of users, are never in a position to *guarantee* trust for a machine learning model.

We believe that the only way forward is through a more powerful third-party auditor that crucially as *more access to the machine learning model,* as this could potentially break the fundamental challenges posed by small explainability regions. We believe that investigating the precise types of access this would entail as an important direction for future work that might have broad practical consequences.

Finally, although our definition of local explainers encompasses several widely used explanation methods, we do note that there are notable exceptions such as Shap (Lundberg and Lee, 2017), which does not fit into our paradigm. As a consequence, one important direction for future work is expanding our framework to encompass other local explanation methods and examine to what degree they can be audited.

## Acknowledgements

This work has been supported by the German Research Foundation through the Cluster of Excellence "Machine Learning - New Perspectives for Science" (EXC 2064/1 number 390727645) and the Carl Zeiss Foundation through the CZS Center for AI and Law.

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

# A Proof of Theorem 4.1

## A.1 An additional assumption

We also include the assumption that the locality parameter be small compared to the tolerance parameters. More precisely, we assume that $\lambda < \epsilon_2^2$.

We believe this to be an *extremely* mild assumption considering that we typically operate in the regime where $\lambda$ is exponentially small with the dimension, $d$, whereas the tolerance parameters are typically between $10^{-2}$ and $10^{-3}$.

## A.2 Main Proof

*Proof.* Fix $\epsilon_1, \epsilon_2, \gamma, \lambda$, and $\mu$, as given in the theorem statement. Our goal is to show the existence of classifier $f$ and explainer $E$ so that the auditor, $A$, is likely to incorrectly estimate the parameter, $L_\gamma(E, f)$. To do so, our strategy will be instead to consider a distribution over choices of $(E, f)$, and show that in expectation over this distribution, $A$ estimates $L_\gamma(E, f)$ poorly.

To this end, we define the following quantities:

1. Let $E$ be the explainer given in Section A.4.

2. Let $f^*$ be the random classifier defined in Section A.4.

3. Let $n$ be any integer with
$$n \leq \frac{1}{2592\lambda^{(1-8max(\epsilon_1,\epsilon_2))}}.$$

4. Let $X$ be a random variable for a set of points $\{x_1, \ldots, x_n\}$ drawn i.i.d from $\mu$.

5. Let $Y = f^*(X)$ be a random variable for $\{f^*(x_1), f^*(x_2), \ldots, f^*(x_n)\}$. $Y$ has randomness over both $f^*$ and $X^*$.

6. Let $\Delta^n = (\mathbb{R}^d)^n \times \{\pm 1\}^n$, and $\sigma$ denote the measure over $\Delta^n$ induced by $(X, Y)$.

7. By definition $E$'s output is *independent* of function, $f^*$. Thus, we will abbreviate $A$'s output by writing
$$A(X, f^*(X), E(f^*, X)) = A(X, Y, E).$$
   This emphasizes that both $X$ and the output of $E$ are independent of $f^*$.

8. We let $I^*$ denote the interval that the auditor seeks to output an estimate it. That is,
$$I^* = \left[ L_{\gamma(1+\epsilon_1)}(E, f^*) - \epsilon_2, L_{\gamma(1-\epsilon_1)}(E, f^*) + \epsilon_2 \right].$$

Using this notation, we seek to lower bound that the auditor fails meaning we seek to lower bound,
$$\Pr_{f^*, X} \left[ A(X, Y, E) \notin I^* \right].$$

To do so, let $T_1$ denote the event
$$T_1 = \mathbb{1} \left( \frac{1}{2} - \epsilon_2 < L_{\gamma(1+\epsilon_1)}(E, f^*) \leq L_{\gamma(1-\epsilon_1)}(E, f^*) < \frac{1}{2} + \epsilon_2 \right),$$

and $T_2$ denote the event
$$T_2 = \mathbb{1} \left( \frac{1}{2} + 3\epsilon_2 < L_{\gamma(1+\epsilon_1)}(E, f^*) \leq L_{\gamma(1-\epsilon_1)}(E, f^*) < \frac{1}{2} + 5\epsilon_2 \right).$$

The key observation is that any estimate, $A(X, Y, E)$, can be inside at most one of the intervals, $[\frac{1}{2} - \epsilon_2, \frac{1}{2} + \epsilon_2]$ and $[\frac{1}{2} + 3\epsilon_2, \frac{1}{2} + 5\epsilon_2]$. Using this, we can re-write our desired probability through the following integration.

Let $(x, y)$ denote specific choices of $(X, Y)$. Note that in this context, $x$ represents a set of points in $(\mathbb{R}^d)^n$, and $y$ represents a set of labels in $\{\pm 1\}^n$. We then have the following:

$$\Pr_{f^*, X}[A(X, Y, E) \notin I^*] = \int_{\Delta^n} \Pr_{f^*}[A(x, y, E) \notin I^* | X = x, Y = y] \, d\sigma(x, y)$$

$$\geq \int_{\Delta^n} \Pr_{f^*}[T_1 | X = x, Y = y] \mathbb{1}\left(A(x, y, E) \notin I_1\right) +$$

$$\Pr_{f^*}[T_2 | X = x, Y = y] \mathbb{1}\left(A(x, y, E) \notin I_1\right) d\sigma(x, y)$$

$$\geq \int_{\Delta^n} \min\left(\Pr_{f^*}[T_1 | X = x, Y = y], \Pr_{f^*}[T_2 | X = x, Y = y]\right) d\sigma(x, y),$$

where the last equation holds because at least one of the events, $A(x, y, E) \notin I_1$ and $A(x, y, E) \notin I_2$, must hold. To bound this last quantity, we utilize Lemma A.9.

Let

$$S^* = \left\{(x, y) : \Pr[T_1 | X = x, Y = y], \Pr[T_0 | X = x, Y = y] \geq \frac{2}{5}\right\}.$$

By Lemma A.9, we have that $\sigma(S^*) \geq \frac{5}{6}$. It follows that

$$\Pr_{f^*, X}[A(X, Y, E) \notin I^*] \geq \int_{\Delta^n} \min\left(\Pr_{f^*}[T_1 | X = x, Y = y], \Pr_{f^*}[T_2 | X = x, Y = y]\right) d\sigma(x, y)$$

$$\geq \int_{S^*} \min\left(\Pr_{f^*}[T_1 | X = x, Y = y], \Pr_{f^*}[T_2 | X = x, Y = y]\right) d\sigma(x, y)$$

$$\geq \int_{S^*} \frac{2}{5} d\sigma(x, y)$$

$$= \frac{2}{5}\sigma(S^*) \geq \frac{1}{3},$$

which completes the proof as this implies with probability at least $\frac{1}{3}$, the Auditors estimate is *not* sufficiently accurate.

$\square$

## A.3  Non-degenerate Distributions

Theorem 4.1 includes the assumption that $\mu$ is *non-degenerate*, which is defined as follows.

**Definition A.1.** We say that data distribution $\mu$ over $\mathbb{R}^d$ is **non-degenerate** if for all $x \in \mathbb{R}^d$, there exists $1 \leq i \leq d$ such that

$$\mu\left(\{x' : x'_i = x_i\}\right) = 0.$$

Being non-degenerate essentially means that at any point, $x$, the data distribution $\mu$ has a finite probability density with respect to *some* feature.

This condition any distribution with a well-defined density over $\mathbb{R}^d$ (i.e. such as a Gaussian) and is also met for most practical data-sets in which any of the features is globally continuously distributed (i.e. mass in kg over a distribution of patients).

We exclude data distributions with point masses because they can pose particularly simple cases in which there is a strict lower bound on how small the local region assigned to a given point can be. For example, in the extreme case where $\mu$ is concentrated on a single point, auditing any model or explanation over $\mu$ is trivial.

We now show a useful property of non-degenerate distributions.

**Lemma A.2.** *Let $\mu$ be a non-degenerate distribution and $R$ be a hyper-rectangle. Then $R$ can be partitioned into two hyper-rectangles, $R_1, R_2$ such that $\mu(R_1), \mu(R_2) \geq \frac{\mu(R)}{4}$.*

*Proof.* Let $R = (a_1, b_1] \times (a_2, b_2] \times \cdots \times (a_d, b_d]$. First, suppose that there exists $1 \leq i \leq d$ such that for all $r \in (a_i, b_i]$,

$$\mu\left(\{x : x = r\} \cap R\right) \leq \frac{\mu(R)}{4}.$$

Let

$$r^* = \sup\left\{r : \mu\left(R \cap \{x : x_i \leq r\}\right) \leq \frac{\mu(R)}{4}\right\}.$$

It follows that setting $R_1 = R \cap \{x : x_i \leq r^*\}$ and $R_2 = R \setminus R_1$ will suffice as $R_1$ will have probability mass at least $\frac{\mu(R)}{4}$ and probability mass at most $\frac{\mu(R)}{2}$.

Otherwise, suppose that no such $i$ exists. Then, thus, there exists $r_1, r_2, \ldots, r_d\}$ such that $\mu(R \cap \{x : x_i = r_i\}) > 0$. It follows that the point $(r_1, \ldots, r_d)$ violates Definition A.1, which is a contradiction. Thus some $i$ exists, which allows us to apply the above argument, finishing the proof. □

## A.4   Constructing $f^*$ and $E$

We begin by partitioning the support of $\mu$ into hyper-rectangles such that each rectangle has probability mass in the interval $[\frac{\alpha}{4}, \alpha]$. We then further partition these rectangles into a large number of equal parts. Formally, we have the following:

**Lemma A.3.** *Let $\alpha > 0$ be fixed, and $K > 0$ be any integer. Then for some integer $L > 0$, there exists a set of hyper-rectangles, $\{R_i^j : 1 \leq i \leq L, 1 \leq j \leq K\}$ such that the following hold:*

1.  *$R_i^1, \ldots R_i^K$ partition rectangle $R_i$.*

2.  *For all $1 \leq i \leq L$, $\alpha \leq \mu(R_i) \leq 4\alpha$.*

3.  *For all $1 \leq i \leq L$ and $1 \leq j \leq K$, $\frac{\mu(R_i)}{4K} \leq \mu(R_i^j) \leq \frac{\mu(R_i)}{K}$.*

*Proof.* First construct $R_1, \ldots, R_L$ by using the following procedure:

1.  Begin with the set $\mathcal{A} = \{R^*\}$ where $R^*$ is a large rectangle containing the support of $\mu$.

2.  If $\mathcal{A}$ contains a rectangle, $R$, such that $\mu(R) > 4\alpha$, then apply Lemma A.2 to split $R$ into two rectangles with mass at least $\frac{\mu(R)}{4}$ and mass at most $\frac{3\mu(R)}{4}$.

3.  Repeat step 2 until no such rectangles, $R$, exist.

This process clearly must terminate in a set of rectangles each of which has mass in the desired range, and also must terminate as a single rectangle can only be cut at most $\frac{\log \frac{1}{\alpha}}{\log \frac{3}{4}}$ times.

Next, to construct $R_i^1, R_i^2, R_i^K$, we simply utilize an analogous procedure, this time starting with $\{R_i\}$ and replacing $\alpha$ with $\frac{\mu(R_i)}{4K}$. □

We now construct a fixed explainer, $E$.

**Definition A.4.** Let $E$ denote the explainer so that for all $x \in supp(\mu)$, we have $E(x) = (R_x, g^{+1})$ where $R_x$ is the unique hyper-rectangle, $R_i$ that contains $x$, and $g^{+1}$ is the constant classifier that always outputs $+1$.

We now construct a distribution over functions $f$, and let $f^*$ be a random function that follows this distribution. We have the following:

**Definition A.5.** Let $f^*$ be a random classifier mapping $\mathbb{R}^d$ to $\{\pm 1\}$ constructed as follows. Let $m$ be an integer and $0 \leq p_1, \ldots, p_{2m}, q_1, \ldots, q_{2m} \leq 1$ be real numbers that satisfy the conditions set forth in Lemma A.10. Then $f^*$ is constructed with the following steps:

1.  Let $P$ be a binary event that occurs with probability $\frac{1}{2}$.

2.  If $P$ occurs, then set $r_i = p_i$ for $1 \leq i \leq p_{2m}$. Otherwise set $r_i = q_i$.

3.  If $x \notin \cup_{i=1}^L R_i$, then $f^*(x) = +1$.

4.  For each rectangle $R_i$, randomly select $1 \leq k \leq 2m$ at uniform.

5. For each sub-rectangle $R_i^j$, with probability $r_k$, set $f(x) = -1$ for all $x \in R_i^j$, and with probability $1 - r_k$, set $f^*(x) = +1$ for all $x \in R_i^j$.

Note that $m$ is constructed based on $\epsilon_1, \epsilon_2$, and $\gamma$ which we assume to be provided as in the statement of Theorem 4.1.

## A.5 Properties of $f^*$ and $E$

We now prove several useful properties of this construction. To do so, we use the following notation:

1. We let $f^*$ denote the random variable representing the way $f$ is generated. We use $f^* = f$ to denote the event that $f^*$ equals a specific function $f : \mathbb{R}^d \to \{\pm 1\}$.

2. We let $P$ denote the indicator variable for the binary event used in Section A.4 to construct $f$.

3. We let $m$ denote the integer from Lemma A.10 that is used to construction $f^*$.

4. We let $X = (x_1, \ldots, x_n) \sim \mu^n$ denote a random variable of $n$ i.i.d selected points from $\mu$. We use $x$ to denote a specific instance of $X$.

5. We let $Y = (y_1, \ldots, y_n)$ be a random variable over labels constructed by setting $y_i = f^*(x_i)$. We similarly use $y$ to denote a specific instance of $Y$.

6. We let $\sigma$ denote the measure over $\left(\mathbb{R}^d \times \{\pm 1\}\right)^n$ associated with $(X, Y)$.

7. We let $\Delta^n$ denote the domain of the pair of random vectors $(X, Y)$ (as done in Section A.2)

We begin with a bounds on the probability that we see any rectangle that has a large number of points selected from it.

**Lemma A.6.** *Let $R_1, \ldots, R_L$ be as defined in section A.4, and $m$ as given. Let $U$ denote the subset of $\Delta^n$ such that*

$$U = \{(x, y) : \exists 1 \leq i \leq z, |X \cap R_i| \geq 2m\}.$$

*Then $\sigma(U) \leq \frac{1}{180}$.*

*Proof.* We bound the probability that a single rectangle, $R_i$, contains at least $2m$ points from $X$, and then apply a union bound over all $L$ rectangles. By construction, $\mu(R_i) \leq 4\lambda$, which implies that for each point $x_j \in X$ the probability that $X_j$ falls within rectangle $R_i$ is at most $4\lambda$. Thus, for any set of $2m$ distinct points from $X$, the probability that they *all* fall within $R_i$ is at most $(4\lambda)^{2m}$. By taking a union bound over all $\binom{n}{2m}$ subsets of $2m$ point from $X$, and substituting our assumed upper bound for $n$ (point 3. of Section A.2), we have the following

$$\begin{aligned}
\Pr[|X \cap R_i| \geq 2m] &\leq \binom{n}{2m} (4\lambda)^{2m} \\
&\leq \left(\frac{en}{2m}\right)^{2m} (4\lambda)^{2m} \\
&\leq \left(\frac{e}{2m} \frac{1}{2592 \max(\epsilon_1, \epsilon_2)\lambda^{1-8\max(\epsilon_1,\epsilon_2)}}\right)^{2m} (4\lambda)^{2m} \\
&= (4\lambda) \left(\frac{e}{2m} \frac{4^{1-\frac{1}{2m}} \lambda^{1-\frac{1}{2m}}}{2592 \max(\epsilon_1, \epsilon_2)\lambda^{1-8\max(\epsilon_1,\epsilon_2)}}\right)^{2m}.
\end{aligned}$$

By definition (see Lemma A.10), $m \geq \frac{1}{16 \max(\epsilon_1, \epsilon_2)}$. Substituting this, and noting that $\lambda^{1-\frac{1}{2m}}$ is increasing with respect to $m$ (since $\lambda < 1$), we have

$$\Pr[|X \cap R_i| \geq 2m] \leq (4\lambda) \left( \frac{e}{2m} \frac{4^{1-\frac{1}{2m}} \lambda^{1-\frac{1}{2m}}}{2592 \max(\epsilon_1, \epsilon_2) \lambda^{1-8\max(\epsilon_1, \epsilon_2)}} \right)^{2m}$$

$$\leq (4\lambda) \left( \frac{e 8 \max(\epsilon_1, \epsilon_2)}{1} \frac{4\lambda^{1-8\max(\epsilon_1, \epsilon_2)}}{2592 \max(\epsilon_1, \epsilon_2) \lambda^{1-8\max(\epsilon_1, \epsilon_2)}} \right)^{2m}$$

$$\leq (4\lambda) \left( \frac{96}{2592} \right)^{2m}$$

$$< \frac{\lambda}{180}.$$

Finally, we apply a union bound over all rectangles. Observe that there are at most $\frac{1}{\lambda}$ such rectangles because by construction each rectangle has mass at most $\lambda$. Thus, our total probability is at most $\frac{1}{\lambda}\frac{\lambda}{180}$ which is at most $\frac{1}{180}$ as desired. □

Next, we leverage the properties from the construction of $f$ to bound the conditional probability of $P = 1$ when $(x, y) \notin U$.

**Lemma A.7.** *Let $(x, y)$ be in the support of $\sigma$ so that $(x, y) \notin U$. Then*

$$\Pr[P = 1|(X, Y) = (x, y)] = \Pr[P = 0|(X, Y) = (x, y)] = \frac{1}{2}.$$

*Proof.* Our main idea will be to use Bayes-rule, and show that $\Pr[(X, Y) = (x, y)|P = 1] = \Pr[(X, Y) = (x, y)|P = 0]$. This will suffice due to the fact that the prior distribution for $P$ is uniform over $\{0, 1\}$. To do so, we first note that $X$ is independent from $P$. For this reason, it suffices to show that

$$\Pr[Y = y|P = 1, X = x] = \Pr[Y = y|P = 0, X = x].$$

To do so, we will express these probabilities in terms of the real numbers, $p_1, \ldots, p_{2m}$ and $q_1, \ldots, q_{2m}$ from which they were constructed (see Definition A.5).

For each rectangle, $R_i$ (see Lemma A.3), let $Y \cap R_i$ denote the function values of all points in the set $X \cap R_i$. It follows from step 4 of Definition A.5 that the values in $Y \cap R_i$ and $Y \cap R_j$ are *independent* from each other. Thus, we can re-write our desired probability as

$$\Pr[Y = y|P = 1, X = x] = \prod_{i=1}^{L} \Pr\left[(Y \cap R_i) = (y \cap R_i)|P = 1, (X \cap R_i) = (x \cap R_i)\right].$$

We now analyze the latter quantity for a rectangle, $R_i$. For convenience, let us relabel indices so that $x \cap R_i = \{x_1, x_2, \ldots, x_l\}$ and $y \cap R_i = \{y_1, \ldots, y_l\}$ for some integer $l \geq 0$. We also let $X_1, \ldots, X_l$ and $Y_1, \ldots, Y_l$ denote the corresponding values for $X \cap R_i$ and $Y \cap R_i$.

We now further assume that that for all $x_a, x_b \in \{x_1, \ldots, x_l\}$, that $x_a$ and $x_b$ are contained within *different* sub-rectangles, $R_i^a, R_i^b$ (see Definition A.5). If this isn't the case, observe that we can *simply remove* the pair $(x_b, y_b)$, as by the construction of $f^*$, this will be forced to be identical to $(x_a, y_a)$.

By applying this assumption, we now have that for a given choice of the parameter $r_k$ (step 4 of Definition A.5), the values of $y_1, \ldots, y_l$ are *mutually independent*. Utilizing this, we have

$$\Pr\left[(Y \cap R_i) = (y \cap R_i)|P = 1, (X \cap R_i) = (x \cap R_i)\right] = \frac{1}{2m} \sum_{j=1}^{2m} \prod_{k=1}^{l} \left( \frac{y_k}{2} - y_k p_j + \frac{1}{2} \right)$$

$$= \frac{1}{2m} \sum_{j=1}^{2m} F(p_j),$$

Where $F$ is a polynomial of degree $l$. Here, the expression, $\frac{y_k}{2} - y_k p_j + \frac{1}{2}$ simply evaluates to $p_j$ if $y_k = -1$ (as $p_j$ is the probability of observing a $-1$) and $1 - p_j$ otherwise.

Next, observe that the only difference when performing this computation for $P = 0$ is that we use the real numbers, $q_1, \ldots q_{2m}$ instead. Thus, we have,

$$\Pr\left[(Y \cap R_i) = (y \cap R_i)|P = 0, (X \cap R_i) = (x \cap R_i)\right] = \frac{1}{2m} \sum_{j=1}^{2m} \prod_{k=1}^{l} \left(\frac{y_k}{2} - y_k q_j + \frac{1}{2}\right)$$

$$= \frac{1}{2m} \sum_{j=1}^{2m} F(q_j),$$

To show these two expression are equal, by assumption $(x, y) \notin U$ which implies that $l < 2m$. Furthermore, by Lemma A.10, $\sum_{k=1}^{2m} p_k^t = \sum_{k=1}^{2m} q_k^t$, for all $0 \leq t \leq l$. It follows that $\sum_{k=1}^{2m} F(p_k) = \sum_{k=1}^{2m} F(q_k)$, which implies our desired result.

$\square$

Next, we bound the probability of events related to the value of $L_\gamma$, the parameter that the Auditor seeks to estimate.

**Lemma A.8.** *Let $T_1$ denote the event that*

$$\frac{1}{2} - \epsilon_2 < L_{\gamma(1+\epsilon_1)} \leq L_{\gamma(1-\epsilon_1)} < \frac{1}{2} + \epsilon_2.$$

*Let $T_0$ denote the event that*

$$\frac{1}{2} + 3\epsilon_2 < L_{\gamma(1+\epsilon_1)} < L_{\gamma(1-\epsilon_1)} \leq \frac{1}{2} + 5\epsilon_2.$$

*Then taken over the randomness of the entire construction, $\Pr[T_1, P = 1], \Pr[T_0, P = 0] \geq \frac{89}{180}$, where $P$ is the binary event defined above.*

*Proof.* By definition, $\Pr[P = 1] = \Pr[P = 0] = \frac{1}{2}$. Thus, it suffices to show that $\Pr[T_1|P = 1], \Pr[T_0|P = 0] \geq \frac{89}{90}$.

We begin with the case that $P = 1$ (the case for $P = 0$ will be similar). For each rectangle, $R_i$, let $r(R_i)$ denote the choice of $r_k$ made for $R_i$ in step 5 of Definition A.5. The crucial observation is that the value of $r(R_i)$ *nearly determines* the local loss that $E$ pays for points in $R_i$ with respect to $f^*$. In particular, if the number of sub-rectangles, $K$, is sufficiently large, then by the law of large numbers, we have that with high probability over the choice of $f^*$, for all rectangles $R_i$ and for all $x \in R_i$,

$$|L(E, f^*, x) - r(R_i)| < 0.01\gamma(\epsilon). \tag{1}$$

Let us fix $K$ to be any number for which this holds, and assume that this value of $K$ is set throughout our construction.

Next, recall by Lemma A.10 that

$$p_1 < p_2 < \cdots < p_m < \gamma(1 - 2\epsilon_1) < \gamma(1 + 2\epsilon_1) < p_{m+1} < \cdots < p_{2m}.$$

Recall that $r(R_i)$ is chosen at uniform among $\{p_1 \ldots p_{2m}\}$ (step 5 of Definition A.5). It follows from Equation 1 that for any $x \in R_i$, and for any $\alpha \in \{\gamma(1 - \epsilon_1), \gamma(1 + \epsilon_1)\}$ that

$$\Pr_{f^*}[L(E, f^*, x) \geq \alpha \text{ for all } x \in R_i] = \frac{1}{2}.$$

Furthermore, because we are conditioning on $P = 1$, the value of $f^*$ within each rectangle, $R_i$, is independent. This implies that we can bound the behavior of $L_\alpha(E, f^*)$ by expressing as a sum of independent variables.

Let $\alpha \in \{\gamma(1 - \epsilon_1), \gamma(1 + \epsilon_1)\}$, we have by Hoeffding's inequality that

$$\Pr\left[L_\alpha(E, f^*) \in \left[\frac{1}{2} - \epsilon_2, \frac{1}{2} + \epsilon_2\right]\right] = \Pr\left[\left(\sum_{i=1}^{L} \mu(R_i)\mathbb{1}\left(L(E, f^*, x) \geq \alpha \text{ for all } x \in R_i\right)\right) \in \left[\frac{1}{2} - \epsilon_2, \frac{1}{2} + \epsilon_2\right]\right]$$

$$\geq 1 - 2\exp\left(-\frac{2\epsilon_2^2}{\sum_{i=1}^{L} \mu(R_i)^2}\right)$$

$$\geq 1 - 2\exp\left(-\frac{2\epsilon_2^2}{16\lambda}\right)$$

$$\geq 1 - \frac{1}{180} = \frac{179}{180};$$

The penultimate inequality holds since $\mu(R_i) \leq 4\lambda$ for each $R_i$, and because there are at most $\frac{1}{\lambda}$ such rectangles. The last inequality holds because $\lambda < \epsilon_2^2$ by the assumption in Section A.1.

Thus by taking a union bound over both values of $\alpha$, we have that $L_\alpha(E, f^*) \in \left[\frac{1}{2} - \epsilon_2, \frac{1}{2} + \epsilon_2\right]$ with probability at least $\frac{89}{90}$. This completes our proof for the case $P = 1$.

For $P = 0$, we can follow a nearly identical argument. The only difference is that the values of $q$ (see Lemma A.10) are selected so that

$$\Pr_{f^*}[L(E, f^*, x) \geq \alpha] \geq \frac{1}{2} + 4\epsilon_2.$$

This results in the expected loss falling within a different interval, and an identical analysis using Hoeffding's inequality gives the desired result. $\square$

The main idea of proving Theorem 4.1 is to show that for many values of $X, Y$, the conditional probabilities of $T_1$ and $T_0$ occurring are both fairly large. This, in turn, will cause the Auditor to have difficulty as its estimate will necessarily fail for at least one of these events.

To further assist with proving this, we have the following additional lemma.

**Lemma A.9.** *Let $S^*$ denote the subset of $\left(\mathbb{R}^d \times \{\pm 1\}\right)^n$ such that*

$$S^* = \left\{(x, y) : \Pr[T_1|(X, Y) = (x, y)], \Pr[T_0|(X, Y) = (x, y)] \geq \frac{2}{5}\right\}.$$

*Then $\sigma(S^*) \geq \frac{5}{6}$.*

*Proof.* Let $S_1' = \{(x, y) : \Pr[T_1|(X, Y) = (x, y)] < \frac{2}{5}\}$, and similarly $S_2' = \{(x, y) : \Pr[T_2|(X, Y) = (x, y)] < \frac{2}{5}\}$. Then $S^* = \left(\mathbb{R}^d \times \{\pm 1\}\right)^n \setminus (S_1' \cup S_2')$. Thus it suffices to upper bound the mass of $S_1'$ and $S_2'$. To do so, let $U$ be the set defined in Lemma A.6. Then we have

$$\frac{89}{180} \leq \Pr[T_1, P = 1]$$

$$= \int_{(\mathbb{R}^d \times \{\pm 1\})^n} \Pr[T_1, P = 1|(X, Y) = (x, y)]d\sigma$$

$$\leq \int_{S_1'} \Pr[T_1, P = 1|(X, Y) = (x, y)]d\sigma + \int_{U \setminus S_1'} \Pr[T_1, P = 1|(X, Y) = (x, y)]d\sigma$$

$$\quad + \int_{\Delta^n \setminus (S_1' \cup U)} \Pr[T_1, P = 1|(X, Y) = (x, y)]d\sigma$$

$$< \frac{2}{5}\sigma(S_1') + (\sigma(U) - \sigma(U \cap S_1')) + \frac{1}{2}\left(\sigma(\Delta^n \setminus U) - \sigma\left((\Delta^n \setminus U) \cap S_1'\right)\right)$$

$$\leq \left(\frac{2}{5} - \frac{1}{2}\right)\sigma(S_1') + \frac{1}{2}\sigma(\Delta^n \setminus U) + \sigma(U)$$

$$\leq \frac{1}{2}\frac{179}{180} + \frac{1}{180} - \frac{\sigma(S_1')}{10}$$

Here we are simply leveraging the fact that $\Pr[P = 1 | X, Y = x, y]$ is precisely $\frac{1}{2}$ when $x, y$ are not in $U$, and consequently that the probability of $T_1$ and $P = 1$ is at most $\frac{2}{5}, 1$, and $\frac{1}{2}$ when $(x, y)$ is in the sets $S_1', U \setminus S_1'$ and $(\Delta^n \setminus U) \setminus S_1'$ respectively. Finally, simplifying this inequality gives us $\sigma(S_1') \leq \frac{1}{12}$.

A completely symmetric argument will similarly give us that $\sigma(S_2') \leq \frac{1}{12}$. Combining these with a union bound, it follows that $\sigma(S^*) \geq \frac{5}{6}$, as desired. □

## A.6 Technical Lemmas

**Lemma A.10.** *For all* $0 < \gamma, \epsilon_1, \epsilon_2 < \frac{1}{48}$, *there exists* $m > 0$, *and real numbers* $0 \leq p_1, p_2, \ldots, p_{2m}, q_1, \ldots, q_{2m} \leq 1$ *such that the following four conditions hold:*

1. *For all* $0 \leq t \leq 2m - 1$, $\sum_{i=1}^{2m} p_i^t = \sum_{i=1}^{2m} q_i^t$.

2. $p_1 \leq p_2 \leq \cdots \leq p_m < \gamma(1 - 2\epsilon_1) < \gamma(1 + 2\epsilon_1) < p_{m+1} \leq \ldots p_{2m}$.

3. $q_1 \leq q_2 \leq \cdots \leq q_{m-1} < q_m = q_{m+1} = \gamma(1 + 2\epsilon_1) < q_{m+2} \leq \ldots q_{2m}$.

4. $\frac{1}{4\epsilon_2} \geq m \geq \frac{1}{8 \max(2\epsilon_1, \epsilon_2)} + 1$.

*Proof.* Let $l$ denote the largest integer that is strictly smaller than $\frac{1}{8 \max(2\epsilon_1, \epsilon_2)}$, and let $\epsilon = \frac{1}{8l}$. It follows that $\epsilon \geq \max(2\epsilon_1, \epsilon_2)$. Let $P_l$ and $Q_l$ be as defined in Lemma A.11. Let $m = 2l$. Then it follows from the definitions of $m, l$ that

$$m = 2l \leq \frac{1}{4 \max(2\epsilon_1, \max(\epsilon_2)} \leq \frac{1}{4\epsilon_2},$$

which proves the first part of property 4 in Lemma A.10. For the second part, by the definition of $l$, we have $l \geq \frac{1}{8 \max(2\epsilon_1, \epsilon_2)} - 1$. Since $\epsilon_1, \epsilon_2 \leq \frac{1}{48}$, it follows that $l \geq 2$ which implies

$$m = 2l \geq l + 2 \geq \frac{1}{8 \max(2\epsilon_1, \epsilon_2)} + 1.$$

Next, let $p_1, \ldots, p_{4l}$ denote the (sorted in increasing order) roots of the polynomial

$$P_l'(x) = P_l \left( \frac{x - \gamma(1 + 2\epsilon_1)}{2\gamma\epsilon} \right).$$

Since the roots of $P_l$ are explicitly given in Lemma A.11, it follows that the middle two roots of $P_l'(x)$ (which are the values of $p_m$ and $p_{m+1}$) satisfy

$$p_m = \gamma(1 + 2\epsilon_1 - 2\epsilon), p_{m+1} = \gamma(1 + 2\epsilon_1 + 2\epsilon).$$

Because $\epsilon' > \epsilon$, these values clearly satisfy the inequalities given by point 2 in the Lemma statement.

Next, define $q_1, \ldots, q_{4l}$ as the (sorted in increasing order) roots of the polynomial,

$$Q_l'(x) = Q_l \left( \frac{x - \gamma(1 + 2\epsilon_1)}{2\gamma\epsilon} \right).$$

Again using Lemma A.11, we see that

$$q_m = q_{m+1} = \gamma(1 + 2\epsilon_1),$$

which satisfies point 3.

To see that $p_i$ and $q_i$ are indeed in the desired range, we simply note that by substitution, both $p_1$ and $p_2$ must be larger than $\gamma(1 + 2\epsilon_1) - 4l(2\gamma\epsilon)$. However, by definition, $4l(2\gamma\epsilon) = \gamma$. Thus, this quantity is larger than 0 which implies that $p_1$ and $q_1$ are both positive. Because $\gamma < \frac{1}{10}$, a similar argument implies that $p_{2m}$ and $q_{2m}$ are at most 1.

Finally, point 1 follows from the fact that $p_1, \ldots, p_{2m}$ and $q_1, \ldots, q_{2m}$ are the complete sets of roots of two polynomials that have matching coefficients for the first $2m$ coefficients. it follows by basic properties of Newton sums that $\sum p_i^t = \sum q_i^t$ for $0 \leq i \leq 2m - 1$, and this proves point 1. □

**Lemma A.11.** *For any $l > 0$, let*

$$P_l(x) = ((x+1)(x+3)\dots(x+4l-1))\,((x-1)(x-3)\dots(x-4l+1))\,.$$

*Let $Q_l(x) = P_l(x) - P_l(0)$. Then $Q_l$ has $2l - 1$ distinct real roots over the interval $(-4l, -1)$, $2l - 1$ distinct real roots over the interval $(1, 4l)$, and a double root at $x = 0$.*

*Proof.* By symmetry, $P_l'(0) = Q_l'(0) = 0$, and by definition $Q_l(0) = 0$. It follows that $x = 0$ is a double root. Next, fix $1 \le i \le l - 1$. By definition, we have that $P_l(4i - 1) = P_l(4i + 1) = 0$. We also have that

$$P_l(4i) = \prod_{j=1}^{2(l+i)} (2j-1) \prod_{j=1}^{2(l-i)} (2j-1).$$

Meanwhile, we also have that $P_l(0) = \left( \prod_{j=1}^{2} l(2j-1) \right)^2$. By directly comparing terms, it follows that $P_l(i)$ is strictly larger than $P_l(0)$. Thus, by the intermediate value theorem, $Q_l$ must have at least one root in both $(4i - 1, 4i)$ and $(4i, 4i + 1)$. Using a similar argument, we can also show that $Q_l$ has at least one root in $(4l - 1, 4l)$.

Since $P_l$ is an even function, it follows that $Q_l$ is as well which means it symmetrically has roots in the intervals $(-4i, -4i + 1)$ for $1 \le i \le l$ and $(-4i - 1, 4i)$ for $1 \le i \le l - 1$. Taken all together, we have constructed $2(l + l - 1) = 4l - 2$ distinct intervals that each contain a root. Since $Q_l$ also has a double root at $x = 0$, it follows that this must account for all of its roots as $deg(Q_l) = deg(P_l) = 4l$. $\qquad\square$

# B Proof of Theorem 4.2

## B.1 Algorithm description

The main idea of our auditor, simple_audit , is to essentially performs a brute-force auditing where we choose a set of points, $X_1$, and attempt to assess the accuracy of their local explanations by using a a wealth of labeled data, $X_2$, to validate it. Our algorithm uses the following steps (pseudocode given in Algorithm 1).

1. (lines 1 -3) We first partition $X$ based on the tolerance parameters, $\epsilon_1, \epsilon_2, \delta$. $X_1$ will be the set of points that we validate, and $X_2$ will be the set of points we use for validation.

2. (lines 8), For *each* point $x$ in $X_1$, we check whether there are enough points from $X_2$ that fall within its local region, $R_x$, to accurate estimate its local loss.

3. (line 9-13) For each point satisfying the criteria in line 8, we evaluate its empirical local loss and then tally up how many points have a loss that is larger than $\gamma$.

4. (line 17) We output the proportion of points with loss larger than $\gamma$ among all points whose loss we measured.

At a high level, we expect this algorithm to succeed as long as we have enough data in *each* of the local regions induced from points in $X_1$.

## B.2 Notation

We use the following:

1. Let $\delta, \epsilon_1, \epsilon_2, \gamma$ be the tolerance parameters defined in Definition 3.1.
2. Let $\lambda = \Lambda(E, f)$ denote the locality of $E, f$ w.r.t. data distribution $\mu$.
3. Let $X_1$ be the set of points that are specifically being audited.
4. Let $X_2$ be the set of points being used to audit.
5. Let $|X_1| = m$. By definition, $m > \frac{\log \frac{1}{\delta}}{\epsilon_2^2}$.
6. We set $|X_2| = n' = n - m$. By definition, $n' > stuff$.

---

**Algorithm 1** $simple\_audit(X, f(X), E(f, X), \epsilon_1, \epsilon_2, \gamma, \delta)$

---

1: $m \leftarrow \frac{61}{\epsilon_2^2} \log \frac{12}{\delta}$.
2: $k \leftarrow \frac{1}{2\gamma^2 \epsilon_1^2} \log \frac{176}{\epsilon_2 \delta}$.
3: $X_1 \leftarrow \{x_1, \ldots, x_m\}, X_2 \leftarrow X \setminus X_1$.
4: $r', b' \leftarrow 0$.
5: **for** $x_i \in X_1$ **do**
6: $\quad (R_{x_i}, g_{x_i}) = E(x_i, f)$
7: $\quad X_1^i = R_{x_i} \cap X_2$.
8: $\quad$ **if** $|X_1^i| \geq k$ **then**
9: $\quad\quad \hat{L}(E, f, x_i) \leftarrow \frac{1}{|X_1^i|} \sum_{x_j \in X_1^i} \mathbb{1}\left(g_{x_i}(x_j) \neq f(x_j)\right)$
10: $\quad\quad$ **if** $\hat{L}(E, f, x_i) > \gamma$ **then**
11: $\quad\quad\quad r' = r' + 1$.
12: $\quad\quad$ **else**
13: $\quad\quad\quad b' = b' + 1$.
14: $\quad\quad$ **end if**
15: $\quad$ **end if**
16: **end for**
17: Return $\frac{r'}{r'+b'}$.

---

7. For any $x \in \mathbb{R}^d$, we let $E(x) = (R_x, g_x)$ be the local explanation outputted for $x$ by explainer $E$.

We also define the following quantities related to estimating how frequently the local loss outputted by the explainer $E$ is above the desired threshold, $\gamma$.

1. Let $r^* = \Pr_{x \sim \mu}[L(E, f, x) \geq \gamma(1 + \epsilon_1)]$.
2. Let $g^* = \Pr_{x \sim \mu}[\gamma(1 - \epsilon_1) \leq L(E, f, x) \leq \gamma(1 + \epsilon_1)]$.
3. Let $b^* = \Pr_{x \sim \mu}[L(E, f, x) \leq \gamma(1 - \epsilon_1)]$.

Here, $r^*$ denotes the probability that a point has a large local error, $b^*$, the probability a point has a low local error, and $g^*$, the probability of an "in-between" case that is nearby the desired threshold, $\gamma$. By the definition of sample complexity (Definition 3.1), the goal of Algorithm 1 is to output an estimate that is inside the interval, $[r^* - \epsilon_2, r^* + g^* + \epsilon_2]$.

Next, we define $r, g, b$ as versions of these quantities that are based on the sample, $X_1$.

1. Let $r = \Pr_{x \sim X_1}[L(E, f, x) \geq \gamma(1 + \epsilon_1)]$.
2. Let $g = \Pr_{x \sim X_1}[\gamma(1 - \epsilon_1) \leq L(E, f, x) \leq \gamma(1 + \epsilon_1)]$.
3. Let $b = \Pr_{x \sim X_1}[L(E, f, x) \leq \gamma(1 - \epsilon_1)]$.

Observe that while $x$ is drawn at uniform from $X_1$ in these quantities, we still use the true loss with respect toe $\mu$, $L(E, f, x)$, to determine whether it falls into $r, g$ or $b$. Because of this, it becomes necessary to define two more *fully empirical* quantities that serve as estimates of $r$ and $b$ (we ignore $g$ as it will merely contribute to a "margin" in our estimation terms).

1. Let $r' = \Pr_{x \sim X_1}\left[(\Pr_{x' \sim X_2}[g_x(x') \neq f(x')|x' \in R_x] > \gamma) \text{ and } |X_2 \cap R_x| \geq \frac{\log \frac{176}{\epsilon_2 \delta}}{2(\gamma \epsilon_1)^2}\right]$.

2. let $b' = \Pr_{x \sim X_1}\left[(\Pr_{x' \sim X_2}[g_x(x') \neq f(x')|x' \in R_x] \leq \gamma) \text{ and } |X_2 \cap R_x| \geq \frac{\log \frac{176}{\epsilon_2 \delta}}{2(\gamma \epsilon_1)^2}\right]$.

The final estimate outputted by Algorithm 1 is *precisely* $\frac{r'}{r'+b'}$. Thus, our proof strategy will be to show that for sufficiently large samples, $r, g, b$ are relatively accurate estimates of $r^*, g^*, b^*$, and in turn $r'$ and $b'$ are relatively accurate estimates of $r$ and $b$. Together, these will imply that our estimate is within the desired interval, $[r^*, b^*]$.

## B.3 The main proof

*Proof.* (Theorem 4.2) Let

$$n \geq \frac{61}{\epsilon_2^2} \log \frac{12}{\delta} + \frac{\log \frac{176}{\epsilon_2 \delta}}{2\lambda\gamma^2\epsilon_1^2} \log \frac{44 \log \frac{176}{\epsilon_2 \delta}}{\epsilon_2 \delta \gamma^2 \epsilon_1^2}.$$

By ignoring log factors, we see that $n = \tilde{O}\left(\frac{1}{\epsilon_2^2} + \frac{1}{\lambda\gamma^2\epsilon_1^2}\right)$, thus satisfying the desired requirement in Theorem 4.2.

Let $X_1$ and $X_2$ be as in Algorithm 1, and let $m, n'$ denote $|X_1|$ and $|X_2|$ respectively. Directly from Algorithm 1, it follows that ,

$$m = \frac{61}{\epsilon_2^2} \log \frac{12}{\delta}, n' \geq \frac{\log \frac{176}{\epsilon_2 \delta}}{2\lambda\gamma^2\epsilon_1^2} \log \frac{44 \log \frac{176}{\epsilon_2 \delta}}{\epsilon_2 \delta \gamma^2 \epsilon_1^2}.$$

By letting $\epsilon = \frac{\epsilon_2}{7}$, and $k = \frac{\log \frac{16}{\epsilon\delta}}{2(\gamma\epsilon_1)^2}$, we have that $m \geq \frac{1}{2\epsilon^2} \log \frac{12}{\delta}$, and that

$$
\begin{aligned}
n' &\geq \frac{\log \frac{176}{\epsilon_2 \delta}}{2\lambda\gamma^2\epsilon_1^2} \log \frac{44 \log \frac{16}{\epsilon\delta}}{\epsilon_2 \delta \gamma^2 \epsilon_1^2} \\
&= \frac{\log \frac{16}{\epsilon\delta}}{2\lambda\gamma^2\epsilon_1^2} \log \frac{4 \log \frac{16}{\epsilon\delta}}{\epsilon\delta \gamma^2 \epsilon_1^2} \\
&= \frac{k \log \frac{8k}{\delta\epsilon}}{\lambda}.
\end{aligned}
$$

Our bounds on $m, k$ and $n'$ allow us to apply Lemmas B.1 and B.4 along with a union bound to get that the following equations hold simultaneously with probability at least $1 - \delta$ over $X \sim \mu^n$:

$$|r - r^*|.|g - g^*|, |b - b^*| \leq \epsilon, \tag{2}$$

$$r(1 - 2\epsilon) \leq r' \leq r + g + b\epsilon \tag{3}$$

$$b(1 - 2\epsilon) \leq b' \leq r\epsilon + g + b. \tag{4}$$

Recall that our goal is to show that $\frac{r'}{r'+b'} \in [r^* - \epsilon_2, r^* + g^* + \epsilon_2]$ holds with probability at least $1 - \delta$. Thus, it suffices to show that this a simple algebraic consequence of equations 2, 3, and 4. To this end, we have

$$
\begin{aligned}
\frac{r'}{r' + b'} &\overset{(a)}{\geq} \frac{r(1 - 2\epsilon)}{r(1 - 2\epsilon) + r\epsilon + g + b} \\
&\geq \frac{r(1 - 2\epsilon)}{r + b + g} \\
&\geq \frac{r}{r + b + g} - 2\epsilon \\
&\overset{(b)}{\geq} \frac{r^* - \epsilon}{r^* + b^* + g^* + 3\epsilon} - 2\epsilon \\
&= \frac{r^*}{1 + 3\epsilon} - \frac{\epsilon}{1 + 3\epsilon} - 2\epsilon \\
&\geq r^*(1 - 3\epsilon) - \epsilon - 2\epsilon \\
&\geq r^* - 4\epsilon - 2\epsilon \\
&\overset{(c)}{\geq} r^* - \epsilon_2.
\end{aligned}
$$

Here step (a) follows from Equations 3 and 4, (b) from Equation 2, and (c) from the fact that $\epsilon = \frac{\epsilon_2}{11}$.

For the other side of the inequality, we have

$$
\begin{aligned}
\frac{r'}{r' + b'} &\overset{(a)}{\leq} \frac{r + g + b\epsilon}{r + g + b\epsilon + b(1 - 2\epsilon)} \\
&\leq \frac{r + g + b\epsilon}{(r + g + b)(1 - \epsilon)} \\
&\leq \frac{r + g}{(r + g + b)(1 - \epsilon)} + \frac{\epsilon}{1 - \epsilon} \\
&\leq \frac{r + g}{r + g + b} + 2\epsilon + \epsilon(1 + 2\epsilon) \\
&\overset{(b)}{\leq} \frac{r^* + g^* + 2\epsilon}{r^* + g^* + b^* - 3\epsilon} + 3\epsilon + 2\epsilon^2 \\
&= \frac{r^* + g^* + 2\epsilon}{1 - 3\epsilon} + 3\epsilon + 2\epsilon^2 \\
&\overset{(c)}{\leq} (r^* + g^*)(1 + 4\epsilon) + 2\epsilon(1 + 4\epsilon) + 3\epsilon + 2\epsilon^2 \\
&\leq r^* + g^* + 6\epsilon + 8\epsilon^2 + 3\epsilon + 2\epsilon^2 \\
&\overset{(d)}{\leq} r^* + g^* + 7\epsilon + 4\epsilon \\
&\overset{(e)}{\leq} r^* + g^* + \epsilon_2
\end{aligned}
$$

Here step (a) follows from Equations 3 and 4, (b) from Equation 2, (c) from the fact that $\frac{1}{1-3\epsilon} \leq 1 + 4\epsilon$, (d) from $\epsilon = \frac{\epsilon_2}{11} \leq \frac{1}{8}, \frac{1}{2}$, and (e) from the $\epsilon = \frac{\epsilon_2}{11}$. $\qquad\square$

## B.4   Concentration lemmas

In this section, we show several lemmas that allow us to bound the behavior of the random variables $r, g, b, r'$ and $b'$ (defined in section B.2). We also use $m$ and $n'$ as they are defined in Section B.2 to be the sizes of $|X_1|$ and $|X_2|$ respectively. Finally, we also let $\epsilon = \frac{\epsilon_2}{11}$.

We begin by bounding the differences between $r, g, b$ and $r^*, g^*, b^*$.

**Lemma B.1.** *Suppose that* $m \geq \frac{1}{2\epsilon^2} \log \frac{12}{\delta}$. *Then with probability at least* $1 - \frac{\delta}{2}$ *over* $X_1 \sim \mu^m$, *the*

$$
|r - r^*|, |g - g^*|, |b - b^*| \leq \epsilon.
$$

*Proof.* Observe that $r$ is the average of $m$ i.i.d binary variables each of which have expected value $r^*$. It follows by Hoeffding's inequality that

$$
\begin{aligned}
\Pr[|r - r^*| > \epsilon] &\leq 2 \exp\left( \frac{-2(\epsilon m)^2}{m} \right) \\
&\leq 2 \exp\left( -2\epsilon^2 \frac{1}{2\epsilon^2} \log \frac{12}{\delta} \right) \\
&= \frac{\delta}{6}.
\end{aligned}
$$

By an identical argument, we see that the same holds for $\Pr[|g - g^*| > \epsilon]$ and $\Pr[|b - b^*| > \epsilon]$. Applying a union bound over all three gives us the desired result. $\qquad\square$

Next, we show that if $n'$ is sufficiently large, then it is highly likely that for any given point $x$, we observe a large number of points from $X_2$ within the explanation region, $R_x$.

**Lemma B.2.** *Let* $x \in supp(\mu)$, *and let* $k > 0$ *be an integer. Suppose that* $n' \geq \frac{k \log \frac{8k}{\delta\epsilon}}{\lambda}$. *Then with probability at least* $1 - \frac{\delta\epsilon}{8}$ *over* $X_2 \sim \mu^m$, $|R_x \cap X_2| \geq k$.

*Proof.* Partition $X_2$ into $k$ sets, $X_2^1, X_2^2, \ldots, X_2^k$ each of which contain at least $\frac{\log \frac{8k}{\delta\epsilon}}{\lambda}$ i.i.d points from $\mu$. Because each point is drawn independently, we have that for any $1 \leq i \leq k$,

$$\Pr[X_2^i \cap R_x = \emptyset] = \left(1 - \Pr_{x' \sim \mu}[x' \in R_x]\right)^{\frac{\log \frac{8k}{\delta\epsilon}}{\lambda}}$$

$$\leq (1 - \lambda)^{\frac{\log \frac{8k}{\delta\epsilon}}{\lambda}}$$

$$\leq \exp\left(-\log \frac{8k}{\delta\epsilon}\right)$$

$$= \frac{\delta\epsilon}{8k}.$$

Here we are using the definition of $\lambda$ as a lower bound on the probability mass of $R_x$.

Next, we show that if $R_x$ has a sufficient number of points, then it is quite likely for the empirical estimate of the local loss at $x$ to be accurate.

**Lemma B.3.** *Let* $x \in supp(\mu)$, *and let* $k \geq \frac{\log \frac{16}{\epsilon\delta}}{2(\gamma\epsilon_1)^2}$. *Then conditioning on there being at least $k$ elements from $X_2$ in $R_x$, the empirical local loss at $x$ differs from the true local loss by at most $\gamma\epsilon_1$ with probability at least $1 - \frac{\delta\epsilon}{8}$. That is,*

$$\Pr_{X_2 \sim \mu^{n'}}\left[\left|L(E, f, x) - \frac{1}{|X_2 \cap R_x|}\sum_{x' \in X_2 \cap R_x}\mathbb{1}\left(g_x(x') \neq f(x')\right)\right| > \gamma\epsilon_1 \,\Big|\, |X_2 \cap R_x| \geq k\right] \leq \frac{\delta\epsilon}{8}.$$

*Proof.* The key idea of this lemma is that the distribution of $k$ points drawn from $\mu$ conditioned on being in $R_x$ is *precisely* the marginal distribution over which $L(E, f, x)$ is defined. In particular, this means that the points in $X_2 \cap R_x$ can be construed as i.i.d drawn from the marginal distribution of $\mu$ over $R_x$. Given this observation, the rest of the proof is a straightforward application of Hoeffding's inequality. Letting $\hat{L}(E, f, x) = \frac{1}{|X_2 \cap R_x|}\sum_{x' \in X_2 \cap R_x}\mathbb{1}\left(g_x(x') \neq f(x')\right)$ and $K = |X_2 \cap R_x|$, we have

$$\Pr_{X_2 \sim \mu^{n'}}\left[\left|L(E, f, x) - \hat{L}(E, f, x)\right| > \gamma\epsilon_1 \,\Big|\, K \geq k\right] \leq 2\exp\left(-\frac{2(K\gamma\epsilon_1)^2}{K}\right)$$

$$\leq 2\exp\left(-\log \frac{16}{\delta\epsilon}\right)$$

$$= \frac{\delta\epsilon}{8},$$

as desired. $\qquad\square$

It follows by a union bound that the probability that least one of the sets in $\{X_2^i \cap R_x : 1 \leq i \leq k\}$ is empty is at most $\frac{\delta\epsilon}{8}$. Thus with probability at least $1 - \frac{\delta\epsilon}{8}$, all the sets are non-empty which implies that $|R_x \cap X_2| \geq k$, completing the proof. $\qquad\square$

Finally, we use the previous two lemmas to show that $r'$ and $b'$ closely approximate $r$ and $b$.

**Lemma B.4.** *Let* $k \geq \frac{\log \frac{16}{\epsilon\delta}}{2(\gamma\epsilon_1)^2}$, *and suppose that* $n' \geq \frac{k \log \frac{8k}{\delta\epsilon}}{\lambda}$. *Then with probability at least $1 - \frac{\delta}{2}$ over $X_2 \sim \mu^{n'}$, the following equations holds:*

$$r(1 - 2\epsilon) \leq r' \leq r + g + b\epsilon,$$

$$b(1 - 2\epsilon) \leq b' \leq r\epsilon + g + b.$$

*Proof.* We begin by defining subsets of $X_1$ that correspond to $r, g, b, r'$ and $b'$. We have

1. Let $R = \{x \in X_1 : L(E, f, x) \geq \gamma(1 + \epsilon_1)\}$.

2. Let $G = \{x \in X_1 : \gamma(1 - \epsilon_1) \leq L(E, f, x) \leq \gamma(1 + \epsilon_1)\}$.

3. Let $B = \{x \in X_1 : L(E, f, x) \leq \gamma(1 - \epsilon_1)]\}$.

4. Let $R' = \left\{ x \in X_1 : \left(\Pr_{x' \sim X_2}[g_x(x') \neq f(x')|x' \in R_x] > \gamma\right) \text{ and } |X_2 \cap R_x| \geq \frac{\log \frac{176}{\epsilon_2 \delta}}{2(\gamma \epsilon_1)^2} \right\}$.

5. Let $B' = \left\{ x \in X_1 : \left(\Pr_{x' \sim X_2}[g_x(x') \neq f(x')|x' \in R_x] \leq \gamma\right) \text{ and } |X_2 \cap R_x| \geq \frac{\log \frac{176}{\epsilon_2 \delta}}{2(\gamma \epsilon_1)^2} \right\}$.

Observe that $r, g, b, r'$, and $b'$ are the probabilities that $x \sim X_1$ is in the sets $R, G, B, R'$, and $B'$ respectively.

Our strategy will be to use the previous lemmas to bound the sizes of the intersections, $R' \cap R, R' \cap B, B' \cap R, B' \cap B'$. To this end, let $x \in R$ be an arbitrary point. By Lemma B.2, with probability at least $1 - \frac{\delta \epsilon}{8}$ over $X_2 \sim \mu^{n'}$, $x \in R' \cup B'$. Furthermore, by Lemma B.3 (along with the definition of $R$), with probability at most $\frac{\delta \epsilon}{8}$, $x \in B'$. Applying linearity of expectation along with Markov's inequality, we get the following two bounds:

$$\Pr_{X_2}\left[\left|R \cap (X_1 \setminus (R' \cap B')\right| > |R|\epsilon\right] \leq \frac{\mathbb{E}_{X_2}[|R \cap (X_1 \setminus (R' \cap B'))|]}{|R|\epsilon}$$

$$\leq \frac{|R|\frac{\delta \epsilon}{8}}{|R|\epsilon} = \frac{\delta}{8},$$

$$\Pr_{X_2}\left[\left|R \cap B'\right| > |R|\epsilon\right] \leq \frac{\mathbb{E}_{X_2}[|R \cap B'|]}{|R|\epsilon}$$

$$\leq \frac{|R|\frac{\delta \epsilon}{8}}{|R|\epsilon} = \frac{\delta}{8}.$$

Applying an analogous line of reasoning stating with $x \in B$, we also have

$$\Pr_{X_2}\left[\left|B \cap (X_1 \setminus (R' \cap B')\right| > |B|\epsilon\right] \leq \frac{\delta}{8},$$

$$\Pr_{X_2}\left[\left|B \cap R'\right| > |B|\epsilon\right] \leq \frac{\delta}{8}.$$

Applying a union bound, none of these events occur with probability at least $1 - \frac{\delta}{2}$ over $X_2 \sim \mu^{n'}$. Thus, it suffices to show that they algebraically imply the desired inequalities. To this end, suppose none of them hold. Then we have,

$$r' = \frac{|R'|}{|X_1|}$$

$$= \frac{|R' \cap B| + |R' \cap G| + |R' \cap R|}{|X_1|}$$

$$\leq \frac{|B|\epsilon + |G| + |R|}{|X_1|}$$

$$= b\epsilon + g + r,$$

$$r' = \frac{|R'|}{|X_1|}$$

$$= \frac{|R' \cap B| + |R' \cap G| + |R' \cap R|}{|X_1|}$$

$$\geq \frac{0 + 0 + |R| - |R \cap B'| - |R \setminus (B' \cup R')|}{|X_1|}$$

$$\geq \frac{|R| - |R|\epsilon - |R|\epsilon}{|X_1|}$$

$$= r(1 - 2\epsilon).$$

The upper and lower bounds on $b'$ are analogous. $\qquad \square$

# C  Proof of Theorem 5.1

## C.1  Definitions and Notation

**Definition C.1.** Let $\alpha = \frac{1}{3670016d^4}$ and $\beta = \frac{1}{3584d^2}$.

**Definition C.2.** Let $S_1, S_2, S_3$ be three $(d-1)$-spheres centered at the origin with radii $(1-\alpha), 1$, and $1+\beta$ respectively for $0 < \alpha, \beta$.

**Definition C.3.** Let $\mu$ denote the data distribution so that $x \sim \mu$ is selected by first selecting $i \in \{1, 2, 3\}$ at uniform, and then selecting $x$ from $S_i$ at uniform.

**Definition C.4.** Let $f$ denote the classifier $\mathbb{R}^d \to \{\pm 1\}$ such that

$$f(x) = \begin{cases} +1 & ||x||^2 \leq 1 - \frac{\alpha}{2} \\ -1 & 1 - \frac{\alpha}{2} < ||x||^2 \leq 1 + \frac{\beta}{2} \\ +1 & ||x^2|| > 1 + \frac{\beta}{2}. \end{cases}$$

**Definition C.5.** Let $x^*$ be an arbitrary point chosen on $S_3$, and let $g$ be any linear classifier, and $B(a, r)$ be any $L_2$-ball that contains $x^*$.

**Lemma C.6.** *There exists $x \in S_2$ and $0 \leq \theta_1, \theta_2, \theta_3 \leq \pi$ such that*

$$S_1 \cap B(a, r) = C(S_1, x(1-\alpha), \theta_1),$$

$$S_2 \cap B(a, r) = C(S_2, x, \theta_2),$$

$$S_3 \cap B(a, r) = C(S_3, x(1+\beta), \theta_3),$$

*where $C(S, x, \theta)$ denotes the spherical cap of angle $\theta$ centered at $x$ on $(d-1)$-sphere $S$ (see Definition C.18).*

## C.2  Main Proof

We begin by showing that the structure of the data distribution $\mu$ provides significant difficulty for linear classifiers. At a high level, the curvature of the spheres, $S_1, S_2, S_3$, make separating them linearly only possible for small portions of the sphere. We formalize this with the following lemma.

**Lemma C.7.** *Let $\theta \geq \frac{\pi}{4}$. Let $x$ be an arbitrary point on $S_2$, and let $T_1(x, \theta), T_3(x, \theta)$ denote the sets*

$$T_1(x, \theta) = C(S_2, x, \theta) \cup C(S_1, x(1-\alpha), \theta),$$

$$T_3(x, \theta) = C(S_2, x, \theta) \cup C(S_3, x(1+\beta), \theta).$$

*Let $g : \mathbb{R}^d \to \{\pm 1\}$ denote any linear classifier. Then $g$ exhibits a loss of at least $\frac{1}{3}$ over the conditional distribution of $\mu$ restricted to either $T_1$ or $T_3$. That is,*

$$\Pr_{x' \sim \mu} [g(x') \neq f(x') | x' \in T_1(x, \theta)], \Pr_{x' \sim \mu} [g(x') \neq f(x') | x' \in T_3(x, \theta)] \geq \frac{1}{3}.$$

Next, we show that if the local explanation region $B(a, r)$, contains a sufficiently large probability mass, then it also must include a region that takes the form given by $T_1$ or $T_3$ from Lemma C.7.

**Lemma C.8.** *Suppose that $\mu(B(a, r)) \geq 3^{1-d}$. Let $T_1$ and $T_3$ be as defined in Lemma C.7. Then there exist $x \in S_2$ and $\theta \geq \frac{\pi}{4}$ such that at least one of the following hold:*

- $T_1(x, \theta) \subseteq B(a, r)$, *and* $\frac{\mu(T_1(x,\theta))}{\mu(B(a,r))} \geq \frac{1}{2}$.

- $T_3(x, \theta) \subseteq B(a, r)$, *and* $\frac{\mu(T_3(x,\theta))}{\mu(B(a,r))} \geq \frac{1}{2}$.

We are now prepared to prove Theorem 5.1.

*Proof.* (Theorem 5.1) Suppose $B(a, r) \geq 3^{1-d}$. Then by Lemma C.8, there exists $\theta \geq \frac{\pi}{4}$ such that either $T_1(x, \theta)$ or $T_3(x, \theta)$ is a subset of $B(a, r)$ and satisfies the conditions outlined above. Suppose that $T_1(x, \theta) \subseteq B(a, r)$ (the other case is analogous).

Let $g$ be any linear classifier. Then it follows from Lemmas C.7 and C.8 that the loss $g$ incurs over the conditional distribution of $\mu$ over $B(a, r)$ can be bounded as follows:

$$\Pr_{z \sim B(a,r)}[g(z) \neq f(z)] \geq \Pr[z \in T_1(x, \theta)] \Pr[g(z) \neq f(z) | z \in T_1(x, \theta)]$$

$$\geq \frac{1}{2} \frac{1}{3} = \frac{1}{6},$$

which concludes the proof. □

## C.3  Proof of Lemma C.7

We will show that the claim holds for $T_3(x, \theta)$ as the proof for $T_1(x, \theta)$ is nearly identical (as $\alpha < \beta$). Let $w \in \mathbb{R}^d$ be a unit vector and $b \in \mathbb{R}$ be a scalar such that

$$g(z) = \begin{cases} 1 & \langle w, z \rangle \geq b \\ -1 & \langle w, z \rangle < b \end{cases}.$$

Our main strategy will be to find a large set of points within $T_3(x, \theta)$ such that $g(z) = g(z(1 + \beta))$ for all $z$ within this set. This will force $g$ to misclassify either $z$ or $z(1 + \beta)$ which will lead to our desired error bound. To this end, define

$$T^* = \left\{ z \in C(S_2, x, \theta) : g(z) = -1, g(z(1 + \beta)) = +1, |\langle x, z \rangle| \leq \cos \frac{\pi}{8} \right\}.$$

**Lemma C.9.** $\frac{\mu(T^*)}{\mu(C(S_2, x, \theta))} \leq \frac{1}{10}$.

*Proof.* Let $z$ be selected at uniform from

$$C(S_2, x, \theta) \setminus \left( C\left( S_2, x, \frac{\pi}{8} \right) \cup C\left( S_2, -x, \frac{\pi}{8} \right) \right).$$

Note that $z$ definitionally satisfies that $|\langle x, z \rangle| \leq \cos \frac{\pi}{8}$. It suffices to upper bound the probability that $g(z) \neq g(z(1 + \beta))$. Let $C_\phi = \{z : \langle z, x \rangle = \cos \phi\}$. Our main idea is to condition on $z \in C_\phi$, and then integrate over all choices of $\phi$. That is, if we let $\phi$ denote the random variable representing the angle between $x$ and $z$, then

$$\Pr_z[g(z) = -1, g(z(1 + \beta)) = +1] = \mathbb{E}_\phi \Pr_{z|\phi}[g(z) = -1, g(z(1 + \beta)) = +1].$$

We will now bound the latter quantity. Fix any $\phi$, and observe that the conditional distribution, $z|\phi$ can be written as

$$z = x \cos \phi + u \sin \phi$$

where $u$ is a random vector in $R^{d-1}$ that is uniformly distributed over the unit sphere, $S^{d-2} \subseteq R^{d-1}$. Rewriting the condition that $g(z) \neq g(z(1 + \beta))$ in terms of $u$, observe that

$$g(z) = -1, g(z(1 + \beta)) = +1 \implies \langle w, z \rangle \leq b \leq \langle w, z(1 + \beta) \rangle$$

$$\implies \frac{b}{1 + \beta} \leq \langle w, z \rangle \leq b$$

$$\implies \frac{b}{1 + \beta} - \langle x \cos \phi, w \rangle \leq \langle w, u \sin \phi \rangle \leq b - \langle x \cos \phi, w \rangle$$

$$\implies \langle w, u \rangle \in \left[ s, s + \frac{\beta}{\sin \phi} \right],$$

where $s$ is a constant that depends solely on $b, w, x$, and $\phi$. Note that we are using the fact that $|b| \leq (1 + \beta)$ as otherwise $g$ would trivially output the same label over all $z \sim \mu$).

By applying Lemma C.17 along with the fact that (by definition of $\phi$)

$$\frac{\beta}{\sin \phi} \leq \frac{\beta}{\sin \frac{\pi}{8}} \leq \frac{1}{1370d^2},$$

we have that

$$\Pr_u\left[ u \in \left[ s, s + \frac{\beta}{\sin \phi} \right] \right] \leq \frac{1}{10},$$

which implies the desired result. □

**Lemma C.10.** $\frac{\mu\left(C\left(S_2, x, \frac{\pi}{8}\right) \cup C\left(S_2, -x, \frac{\pi}{8}\right)\right)}{\mu(C(S_2, x, \theta))} \leq \frac{7}{30}$.

*Proof.* By symmetry, $\mu\left(C\left(S_2, x, \frac{\pi}{8}\right)\right) = \mu\left(C\left(S_2, -x, \frac{\pi}{8}\right)\right)$ so it suffices to bound one of them. Since $\theta \geq \frac{\pi}{4}$ by assumption, applying Lemma C.20, we have

$$
\begin{aligned}
\frac{\mu\left(C\left(S_2, x, \frac{\pi}{8}\right) \cup C\left(S_2, -x, \frac{\pi}{8}\right)\right)}{\mu\left(C(S_2, x, \theta)\right)} &\leq \frac{2\mu\left(C\left(S_2, x, \frac{\pi}{8}\right)\right)}{\mu\left(C(S_2, x, \theta)\right)} \\
&\leq \frac{2\mu\left(C\left(S_2, x, \frac{\pi}{8}\right)\right)}{\mu\left(C\left(S_2, x, \frac{\pi}{4}\right)\right)} \\
&\leq 2\frac{1}{2}\left(\frac{\sin\frac{\pi}{8}}{\sin\frac{\pi}{4}}\right)^{d-2} \\
&\leq \frac{7}{30},
\end{aligned}
$$

as $d \geq 5$ in the assumption of Theorem 5.1. $\qquad\square$

We are now prepared to prove the main lemma.

*Proof.* (Lemma C.7) Let $A^* \subseteq C(S_2, x, \theta)$ be defined as the set of all points for which $g$ classifies both the point and its image in $(1 + \beta)S_3$ correctly. That is,

$$
A^* = \{z \in C(S_2, x, \theta) : g(z) = -1, g((1 + \beta)z) = +1\}.
$$

By the previous two lemmas, we have

$$
\begin{aligned}
\frac{\mu(A^*)}{\mu(C(S_2, x, \theta))} &\leq \frac{\mu(T^* \cup C\left(S_2, x, \frac{\pi}{8}\right) \cup C\left(S_2, -x, \frac{\pi}{8}\right))}{\mu(C(S_2, x, \theta))} \\
&\leq \frac{1}{10} + \frac{7}{30} = \frac{1}{3}
\end{aligned}
$$

Each $z \in A^*$ is a point for which both $z$ and $(1 + \beta)z$ are correctly classified, and each $z \in C(S_2, x, \theta) \setminus A^*$ corresponds to either $z$ being misclassified, or $(1 + \beta)z$ being misclassified. It follows that the overall accuracy of $g$ over $T_3(x, \theta)$ is at most

$$
\begin{aligned}
\Pr_{z \sim T_3(x,\theta)}[g(z) = f(z)] &\leq \Pr_{z \sim C(S_2,x,\theta)}[z \in A^*] + \frac{1}{2}\Pr_{z \sim C(S_2,x,\theta)}[z \notin A^*] \\
&\leq \frac{1}{2}\left(1 + \Pr_{z \sim C(S_2,x,\theta)}[z \in A^*]\right) \\
&\leq \frac{2}{3}
\end{aligned}
$$

Thus $g$ must incur loss at least $\frac{1}{3}$ over $T_3(x, \theta)$, as desired. $\qquad\square$

## C.4 Proof of Lemma C.8

Throughout this section, we assume that $\mu(B(a, r)) \geq 3^{1-d}$.

**Lemma C.11.** $\max(\theta_1, \theta_2, \theta_3) \geq \frac{\pi}{3}$.

*Proof.* Assume towards a contradiction that this does not hold. Let $x$ be as in Lemma C.6. Then by the Definition of $\mu$ (Definition C.3) and Lemma C.6, it follows that

$$
\begin{aligned}
\mu(B(a, r)) &= \mu(C(S_1, x(1 - \alpha), \theta_1)) + \mu(C(S_2, x, \theta_2)) + \mu(C(S_3, x(1 + \beta), \theta_3)) \\
&= \frac{1}{3}\left(\Psi(\theta_1) + \Psi(\theta_2) + \Psi(\theta_3)\right) \\
&< \Psi\left(\frac{\pi}{3}\right),
\end{aligned}
$$

where $\Psi$ is as defined in Section C.6. However, Lemma C.19 implies that $\Psi\left(\frac{pi}{3}\right) \leq 3^{1-d}\Psi(\pi) = 3^{1-d}$. This contradicts our assumption on $\mu(B(a, r))$ and implies the desired result. $\qquad\square$

**Lemma C.12.** $r \geq 1 - \alpha$.

*Proof.* Lemma C.11 implies that $B(a, r)$ must intersect some sphere among $S_1, S_2, S_3$ in a spherical cap of an angle at least $\frac{\pi}{3}$. Basic geometry implies that $r \geq \min(rad(S_1), rad(S_2), rad(S_3))$ where $rad(S_i)$ denotes the radius of $S_i$. The desired result follows from the fact that $1 - \alpha - rad(S_1) \leq rad(S_2), rad(S_3)$. $\qquad\square$

**Lemma C.13.** $|\theta_2 - \max(\theta_1, \theta_3)| \leq \frac{1}{4d}$.

*Proof.* We first compute $\theta_1, \theta_2, \theta_3$ in terms of $a, r, \alpha$, and $\beta$. We begin with $\theta_2$, and note that the expressions for $\theta_1$ and $\theta_3$ can be similarly derived. To this end, we have

$$
\begin{aligned}
S_2 \cap B(a, r) &= \{x : ||x|| = 1, ||x - a|| \leq r\} \\
&= \left\{x : ||x|| = 1, \langle x, x \rangle - 2\langle x, a \rangle + \langle a, a \rangle \leq r^2\right\} \\
&= \left\{x : ||x|| = 1, \left\langle \frac{x}{||x||}, \frac{a}{||a||} \right\rangle \geq \frac{1 + a^2 - r^2}{2a}\right\},
\end{aligned}
$$

where we use $a$ to denote $||a||$ in a slight abuse of notation.

It follows from Lemma C.6 that

$$
\cos\theta_2 = \frac{1 + a^2 - r^2}{2a}.
$$

We can similarly show that

$$
\cos\theta_1 = \frac{(1 - \alpha)^2 + a^2 - r^2}{2(1 - \alpha)a}, \cos\theta_3 = \frac{(1 + \beta)^2 + a^2 - r^2}{2(1 + \beta)a}.
$$

Let $h : \mathbb{R} \to \mathbb{R}$ be the function defined as $h(s) = \frac{s^2 + a^2 - r^2}{2sa}$. Thus,

$$
\cos\theta_1 = h(1 - \alpha), \cos\theta_2 = h(1), \cos\theta_3 = h(1 + \beta).
$$

Note that in cases where $h$ is outside of the interval $[-1, 1]$ (meaning $\theta_i$ would not be defined), we simply set $\theta_i$ equal to $\pi$ and $0$ respectively, as these quantities still accurately describe the intersection between $B(a, r)$ and the corresponding sphere, $S_i$.

**Case 1:** $0 \leq a \leq \frac{\beta}{2}$
By definition, $B(a, r)$ contains $x^*$ and therefore intersects $S_3$. It follows from the triangle inequality that $r \geq 1 + \frac{\beta}{2}$. However, this implies that $B(a, r)$ must contain the entirety of $S_2$ and $S_1$, which implies that $\theta_1 = \theta_2 = \max(\theta_1, \theta_3) = \pi$, thus implying the lemma statement.

**Case 2:** $\frac{\beta}{2} < a \leq 1 - \alpha$
If $r > 1 + 2\beta$, then $B(a, r)$ will contain $S_1, S_2$ and $S_3$, which implies $\theta_1 = \theta_2 = \theta_3 = \pi$ (implying the lemma statement). Thus, assume $r \leq 1 + 2\beta$.

Differentiating $h$ w.r.t. $s$ gives

$$
h'(s) = \frac{1}{2a}\left(1 + \frac{r^2 - a^2}{s^2}\right).
$$

By Lemma C.12, $r^2 \geq a^2$, which implies that $h'(s)$ is nonnegative for $s \in [1-\alpha, 1+\beta]$. Furthermore, we have that over the interval, $[1 - \alpha, 1 + \beta]$,

$$h'(s) = \frac{1}{2a}\left(1 + \frac{r^2 - a^2}{s^2}\right)$$

$$\leq \frac{1}{\beta}\left(1 + \frac{(1+2\beta)^2 - \left(\frac{\beta}{2}\right)^2}{(1-\alpha)^2}\right)$$

$$= \frac{1}{\beta}\left(1 + \frac{1 + 4\beta + 3.75\beta^2}{(1-\alpha)^2}\right)$$

$$\leq \frac{1}{\beta}\left(1 + \frac{1 + 4(0.25) + 3.75(0.25)^2}{0.875^2}\right)$$

$$\leq \frac{4}{\beta}.$$

This is obtained by substituting appropriate upper and lower bounds for $r, a, s, \alpha$, and $\beta$. Because $h'(s)$ is nonnegative over the interval, we must have that $h(1 - \alpha) \leq h(1) \leq h(1 + \beta)$ which implies $\theta_1 \geq \theta_2 \geq \theta_3$ (as $\cos$ is a decreasing function). It follows from our upper bound on $h'(s)$ that

$$|\cos\theta_2 - \cos(\max(\theta_1, \theta_3))| = \cos(\theta_2) - \cos(\theta_1)$$

$$= h(1) - h(1 - \alpha)$$

$$= \int_{1-\alpha}^{1} h'(s)ds$$

$$\leq \int_{1-\alpha}^{1} \frac{4}{\beta}ds$$

$$= \frac{4\alpha}{\beta}.$$

Applying Lemma C.14 implies that $|\theta_2 - \max(\theta_1, \theta_3)| \leq 8\sqrt{\frac{\alpha}{\beta}} = \frac{1}{4d}$, which implies the lemma statement.

**Case 3:** $a > 1 - \alpha$
First suppose that $|a - r| > 3$. If $r > a + 3$, then the triangle inequality implies that $S_1, S_2, S_3 \subseteq B(a, r)$ which implies the desired result. On the other hand, if $r < a - 3$, then we must have $a > 3$, and that $B(a, r)$ is disjoint from $S_1, S_2, S_3$ which again implies the desired result. Thus, we assume that $|a - r| \leq 3$.

We now use a similar strategy to the previous case, and bound the derivative, $h'(s)$. By substituting that $|a - r| \leq 3$, we have, for $s \in [1 - \alpha, 1 + \beta]$,

$$|h'(s)| = \left|\frac{1}{2a}\left(1 + \frac{r^2 - a^2}{s^2}\right)\right|$$

$$= \left|\frac{1}{2a}\left(1 + \frac{(2a+3)(3)}{s^2}\right)\right|$$

$$= \left|\frac{1}{2a}\left(1 + \frac{(2a+3)(3)}{(1-\alpha)^2}\right)\right|$$

$$= \left|\frac{1}{2a}\left(1 + \frac{(2a+3)(3)}{(0.875)^2}\right)\right|$$

$$\leq \left|\frac{1}{2a}\left(1 + 4(2a+3)\right)\right|$$

$$\leq \left|\frac{1}{2a}\left(13 + 8a\right)\right|$$

$$\leq 4 + 10 = 14.$$

Here we are exploiting the fact that $1 - \alpha \geq \frac{\sqrt{3}}{2}, 0.65$. It follows by the same argument given in Case 2 that $|\cos\theta_2 - \cos(\max(\theta_1, \theta_3))| \leq 14\beta$. Applying Lemma C.14 implies $|\theta_2 - \max(\theta_1, \theta_3)| \leq 4\sqrt{14\beta} = \frac{1}{4d}$, as desired.

$\square$

Now we are ready to prove the lemma.

*Proof.* (Lemma C.8) Let $x$ be as in Lemma C.6, and let $\theta^* = \max(\theta_1, \theta_2, \theta_3)$. Then by applying Lemma C.6 to the Definition of $\mu$ (Definition C.3) gives us

$$\mu\left(B(a, r)\right) = \mu\left(C(S_1, x(1-\alpha), \theta_1)\right) + \mu\left(C(S_2, x, \theta_2)\right) + \mu\left(C(S_3, x(1+\beta), \theta_3)\right)$$
$$= \frac{1}{3}\Psi(\theta_1) + \frac{1}{3}\Psi(\theta_2) + \frac{1}{3}\Psi(\theta_3)$$
$$\leq \Psi(\theta^*).$$

Here $\Psi$ denotes the function defined in Section C.6.

Next, let $\theta = \min(\max(\theta_1, \theta_3), \theta_2)$. Let $T_1(x, \theta)$ and $T_3(x, \theta)$ be as defined in Lemma C.7. Observe that if $\theta_1 \geq \theta_3$, then

$$T_1(x, \theta) \subseteq C(S_1, x(1-\alpha), \theta_1) \cup C(S_2, x, \theta_2) \subseteq B(a, r),$$

and otherwise,

$$T_3(x, \theta) \subseteq C(S_3, x(1+\beta), \theta_3) \cup C(S_2, x, \theta_2) \subseteq B(a, r).$$

Thus, at least one of these sets is part of $B(a, r)$. We now show that these sets have the desired mass. By the definition of $\theta^*$, we have

$$\frac{\mu(T_1(x, \theta))}{\mu(B(a, r))}, \frac{\mu(T_3(x, \theta))}{\mu(B(a, r))} \geq \frac{2\mu(C(S_2, x, \theta))}{3\mu(C(S_2, x, \theta^*))}.$$

Next, Lemma C.11 implies that $\theta^* \geq \frac{\pi}{3}$, and Lemma C.13 implies that $\theta^* - \theta \leq \frac{1}{4d}$. It follows that

$$\theta \geq \theta^* - \frac{1}{4d} \geq \theta^*\left(1 - \frac{1}{4d}\right).$$

Substituting this, we find that

$$\frac{2\mu(C(S_2, x, \theta))}{3\mu(C(S_2, x, \theta^*))} = \frac{\frac{2}{3}\Psi(\theta)}{\Psi(\theta^*)}$$
$$\geq \frac{2}{3}\frac{\Psi\left(\theta^*\left(1 - \frac{1}{4d}\right)\right)}{\Psi(\theta^*)}$$
$$\geq \frac{2}{3}\left(1 - \frac{1}{4d}\right)^{d-1}$$
$$\geq \frac{2}{3}\left(\frac{1}{e}\right)^{1/4}$$
$$\geq \frac{1}{2},$$

where the last steps follow from Lemmas C.19 and C.16. This completes the proof. $\square$

## C.5    Technical Lemmas

**Lemma C.14.** *Suppose $\phi_1, \phi_2 \in [0, \pi]$ such that $|\cos(\phi_1) - \cos(\phi_2)| \leq c$. Then $|\phi_1 - \phi_2| \leq 4\sqrt{c}$.*

*Proof.* WLOG, suppose $\phi_1 \leq \phi_2$. Let $x = \phi_2 - \phi_1$.

Using the sum to product rules, it follows that

$$\alpha \geq |\cos\phi_1 - \cos\phi_2|$$

$$= \left| -2\sin\frac{\phi_1 - \phi_2}{2}\sin\frac{\phi_1 + \phi_2}{2} \right|$$

$$\geq \left| 2\sin\frac{x}{2}\sin\frac{\phi_1 + \phi_2}{2} \right|.$$

However, observe that $\pi - \frac{\phi_1 + \phi_2}{2} \geq \phi_2 - \frac{\phi_1 + \phi_2}{2} = \frac{x}{2}$ and that $\frac{\phi_1 + \phi_2}{2} \geq \frac{0 + 0 + x}{2} = \frac{x}{2}$. It follows that $\frac{\phi_1 + \phi_2}{2} \in [\frac{x}{2}, \pi - \frac{x}{2}]$, which implies that $\sin\frac{\phi_1 + \phi_2}{2} \geq \sin\frac{x}{2}$. Substituting this, we have

$$c \geq \left| 2\sin\frac{x}{2}\sin\frac{\phi_1 + \phi_2}{2} \right|$$

$$\geq 2\sin^2\frac{x}{2}$$

We now do casework based on $x$. First suppose that $x \geq \frac{\pi}{2}$. Then $c \geq 2\sin^2\frac{\pi}{4} = 1$. By definition, $x \leq \pi$, so it follows that $x \leq 4\sqrt{\alpha}$, implying the desired result.

Otherwise, if $x \leq \frac{\pi}{2}$, then $\sin\frac{x}{2} \geq \frac{x}{4}$, as the function $t \mapsto \sin(t) - \frac{t}{2}$ is nonnegative on the interval $[0, \frac{\pi}{2}]$. Substituting this, we see that $c \geq \frac{x^2}{8}$. Thus $x \leq \sqrt{8c} < 4\sqrt{c}$, as desired. □

**Lemma C.15.** *For $0 \leq c \leq 1$ and $0 \leq \theta \leq \pi$, $\sin(c\theta) \geq c\sin(\theta)$.*

*Proof.* Let $f(\theta) = \sin(c\theta) - c\sin(\theta)$. Observe that $f(0) = 0$. Furthermore, for $\theta \in [0, \pi]$, we have $f'(\theta) = c\cos(c\theta) - c\cos(\theta) = c(\cos(c\theta) - \cos(\theta))$. Since $\cos$ is a decreasing function on the interval $[0, \pi]$, it follows that $\cos(c\theta) \geq \cos(\theta)$, which implies $f'(\theta) \geq 0$. Thus $f$ is non-decreasing on the interval, and the desired inequality holds. □

**Lemma C.16.** *For all $x > 1$, $\left(1 - \frac{1}{x}\right)^{x-1} \geq \frac{1}{e}$.*

*Proof.* Let $f(x) = \left(1 - \frac{1}{x}\right)^{x-1}$. It is well known that $\lim_{x \to \infty} f(x) = \frac{1}{e}$ and $\lim_{x \to 1+} f(x) = 1$. Thus it suffices to show that $f(x)$ is a non-increasing function. To do so, we will show that $\ln f(x)$ is non-increasing by taking its derivative. We have

$$\frac{d(\ln f(x))}{dx} = \frac{d}{dx}\left((x-1)\ln\frac{x-1}{x}\right)$$

$$= \frac{d}{dx}\left((x-1)\ln(x-1) - (x-1)\ln x\right)$$

$$= \left(\ln(x-1) + \frac{x-1}{x-1}\right) - \left(\ln(x) + \frac{x-1}{x}\right)$$

$$= \frac{1}{x} - (\ln(x) - \ln(x-1))$$

$$= \frac{1}{x} - \int_{x-1}^{x} \frac{1}{t}dt$$

$$\leq \frac{1}{x} - \int_{x-1}^{x} \frac{1}{x}dt$$

$$= \frac{1}{x} - \frac{1}{x} = 0.$$

□

**Lemma C.17.** *Let $z$ be a point chosen at uniform over $S_2$, and let $w$ be a fixed unit vector. Then if $t \leq \frac{1}{1370d^2}$, then for any $s \in \mathbb{R}$,*

$$\Pr_z[\langle w, z \rangle \in [s, s+t]] \leq \frac{1}{10}.$$

*Proof.* Let $\theta$ denote the random variable that represents the angle between $w$ and $z$. Applying Lemma C.14, it follows that for some choice of $s' \in \mathbb{R}$ that

$$\Pr_z[\langle w, z \rangle \in [s, s+t] \leq \Pr_\theta[\theta \in [s', s' + 4\sqrt{t}].$$

We now bound this quantity by utilizing the quantity $\Psi$ (defined in Section C.6). We have,

$$\begin{aligned}
\Pr_\theta[\theta \in [s', s' + 4\sqrt{t}] &= \frac{\int_{s'}^{s'+4\sqrt{t}} \sin^{(d-2)} \phi d\phi}{\int_0^\pi \sin^{(d-2)} \phi d\phi} \\
&\leq \frac{2 \int_{\frac{\pi}{2}-2\sqrt{t}}^{\frac{\pi}{2}} \sin^{(d-2)} \phi d\phi}{2 \int_0^{\frac{\pi}{2}} \sin^{(d-2)} \phi d\phi} \\
&= 1 - \frac{\Psi\left(\frac{\pi}{2} - 2\sqrt{t}\right)}{\Psi\left(\frac{\pi}{2}\right)}.
\end{aligned}$$

Here we have simply chosen the interval of length $4\sqrt{t}$ that maximizes the corresponding the integral. Next, we continue by applying Lemmas C.19 and C.16 to get

$$\begin{aligned}
\Pr_\theta[\theta \in [s', s' + 4\sqrt{t}] &\leq 1 - \frac{\Psi\left(\frac{\pi}{2} - 2\sqrt{t}\right)}{\Psi\left(\frac{\pi}{2}\right)} \\
&\leq 1 - \left(1 - \frac{4\sqrt{t}}{\pi}\right)^{d-1} \\
&\leq 1 - \left(1 - \frac{1}{29d}\right)^{d-1} \\
&= 1 - \left(\left(1 - \frac{1}{29d}\right)^{29(d-1)}\right)^{1/29} \\
&\leq 1 - \left(\left(1 - \frac{1}{29d}\right)^{29d-1}\right)^{1/29} \\
&\leq 1 - \left(\frac{1}{e}\right)^{1/29} \\
&\leq \frac{1}{10},
\end{aligned}$$

as desired. $\square$

### C.6 Spherical Caps

**Definition C.18.** Let $S$ be a $(d-1)$ sphere centered at the origin, let $0 \leq \theta \leq \pi$ be an angle, and let $x \in S$ be a point. We let $C(S, x, \theta)$ denote the **spherical cap** with angle $\theta$ centered at $x$, and it consists of all points, $x' \in S^{d-1}$, such that $\frac{\langle x, x' \rangle}{||x||||x'||} \geq \cos \phi$.

Here we take the convention of associating $C(S_i, x_i, 0)$ with both the empty set and with $\{x_i\}$. While these are distinct sets, they both have measure 0 under $\pi$. We also associate $C(S_i, x_i, \pi)$ with the entirety of $S_i$.

We let $\Psi(\theta)$ denote the ratio of the $(d-1)$-surface area of the region, $C(S, x, \theta)$, to the $(d-1)$-surface area of the entire sphere. Thus, $\Psi(\theta)$ denotes the fraction of the sphere covered by a spherical cap of angle $\theta$. By standard integration over spherical coordinates, we have

$$\Psi(\theta) = \frac{\int_0^\theta \sin^{(d-2)} \phi d\phi}{\int_0^\pi \sin^{(d-2)} \phi d\phi}.$$

Next, we bound $\Psi(\theta)$ with the following inequality.

**Lemma C.19.** *Let $0 \leq \theta \leq \pi$ and let $0 \leq c \leq 1$. Then*

$$\frac{\Psi(c\theta)}{\Psi(\theta)} \geq c^{d-1}.$$

*Proof.* By applying Lemma C.15 to the definition of $\Psi$, we have the following manipulations.

$$\begin{aligned}
\frac{\Psi(c\phi)}{\Psi(\phi)} &= \frac{\int_0^{c\phi} \sin^{d-2}\theta d\theta}{\int_0^{\phi} \sin^{d-2}\theta d\theta} \\
&= \frac{\int_0^{\phi} \sin^{d-2}(cu)(cdu)}{\int_0^{\phi} \sin^{d-2}\theta d\theta} \\
&\geq \frac{\int_0^{\phi} (c\sin(u))^{d-2}(cdu)}{\int_0^{\phi} \sin^{d-2}\theta d\theta} \\
&= c^{d-1}.
\end{aligned}$$

$\square$

We similarly have an upper bound on this ratio.

**Lemma C.20.** *Let $0 \leq \theta \leq \frac{\pi}{2}$ and $0 \leq c \leq 1$. Then*

$$\frac{\Psi(c\theta)}{\Psi(\theta)} \leq c \left( \frac{\sin c\phi}{\sin \phi} \right)^{d-2}.$$

*Proof.* We similarly have,

$$\begin{aligned}
\frac{\Psi(c\phi)}{\Psi(\phi)} &= \frac{\int_0^{c\phi} \sin^{d-2}\theta d\theta}{\int_0^{\phi} \sin^{d-2}\theta d\theta} \\
&= \frac{\int_0^{\phi} \sin^{d-2}(cu)(cdu)}{\int_0^{\phi} \sin^{d-2}\theta d\theta} \\
&\leq \frac{\int_0^{\phi} \left( \sin(u)\frac{\sin c\phi}{\sin \phi} \right)^{d-2}(cdu)}{\int_0^{\phi} \sin^{d-2}\theta d\theta} \\
&= c \left( \frac{\sin c\phi}{\sin \phi} \right)^{d-2}.
\end{aligned}$$

Here we are using the fact that $t \mapsto \frac{\sin ct}{\sin t}$ is a non-decreasing function for $t \in [0, \pi]$. $\square$

