# OpenReview forum: "Auditing Local Explanations is Hard"
_NeurIPS.cc/2024/Conference — NeurIPS 2024 poster_

### Official Review · Reviewer_mYL9 · 2024-07-10

**Soundness:** 3
**Presentation:** 3
**Contribution:** 3
**Rating:** 7
**Confidence:** 4

**Summary:**

The paper addresses the challenges in verifying the accuracy of local explanations for machine learning models, especially when the model is not fully known and access to it is limited. The primary focus is on minimizing the number of times the model and explainer are accessed during the auditing process. The contributions of the paper are as follows:

**C1. Defining Auditing of Local Explanations:** The paper provides a formal definition of what it means to audit local explanations. It sets the groundwork for systematically evaluating the reliability of these explanations, which are crucial for understanding and trusting machine learning models.

**C2. Importance of the Region’s size**: It highlights that the region to which the local explainer applies is a critical property. Understanding the scope and limits of where the explanation is valid is essential for accurate auditing. This insight helps in identifying when and where an explanation might fail to represent the model correctly.

**C3. Bounds on Auditing Complexity**: The paper establishes both lower and upper bounds on the sample complexity required for auditing local explanations. These bounds are presented as functions of the newly identified property, which is the region’s size. This provides a theoretical framework for understanding the minimal and maximal data requirements for effective auditing.

**Strengths:**

**S1. Framework for Auditing**: It proposes a theoretical framework for auditing local explanations, which is a significant step towards developing more rigorous and reliable methods for verifying the trustworthiness of explanations provided by machine learning models.

**S2. Identification of Key Metrics**: The introduction of the explainability loss function, provides a quantifiable measure for evaluating local explanations for the original model, offering a systematic way to assess explanation quality.

**S3. Highlighting the Importance of Locality**: The analysis which provide upper and lower bounds highlights the importance of the "locality" of explanations, bringing attention to a previously underexplored aspect in the explainability literature.

**Weaknesses:**

**W1. Lack of Evaluation**: The paper does not include an evaluation on real-world datasets. Although the authors suggest that their results could have significant practical implications (“Our results might have far-reaching practical consequences”), an initial step should be to perform evaluations on actual data to validate their findings.

**W2. Limited Discussion with Previous Research**: The paper could benefit from a more thorough discussion of how its findings relate to and build upon previous research in the field. Specifically, the authors mentioned that [Dasgupta 2022] is the most similar to their work, except it is limited to discrete explanations rather than continuous. However, it is not clear whether, in the discrete setting, the proposed work aligns with Dasgupta’s consistency metric, sufficiency metric, or if it does not coincide with any of these previous metrics.
It may also be worth discussing and comparing with the recent work of [Bassan 2023], which suggests a verification method for finding minimal sufficient explanations.

- Dasgupta, S., Frost, N. and Moshkovitz, M., 2022. Framework for evaluating faithfulness of local explanations. In International Conference on Machine Learning
- Bassan, S. and Katz, G., 2023. Towards formal XAI: formally approximate minimal explanations of neural networks. In International Conference on Tools and Algorithms for the Construction and Analysis of Systems

**W3. Focus on a Specific Type of Explanations**: In the paper, the presentation is a bit misleading because it suggests that explanations can be general. However, the focus is on one type of explanation – those that approximate the true model on a region of examples. This is not a general (local) explanation method, even though it includes quite a few types of explanations.

**Questions:**

See Weaknesses.

**Limitations:**

--

---

> ### Author Rebuttal · Authors · 2024-08-06
>
> We thank the reviewer for their thoughtful review. We will respond to each weakness in order.
>
> W1. We agree that empirical validation is an important direction for further work, and will emphasize this in the final version of our paper. We chose to take a theoretical focus because the main result of our work is a lower bound that applies to any algorithm.
>
> W2. We thank the reviewer for the added reference, and will include a more detailed comparison with prior works in the final version of our paper. With regards to Dasgupta et. al., it is indeed correct that the notion of "local consitency" (Definition 1) provided in their paper precisely matches our notion of "local loss" (Definition 2.3) when the region they define, $C_\pi$, is construed as a local explanation region. However, a key difference with their work is that ours provides lower bounds on the amount of data needed to estimate the local loss (over the entire distribution), and this result is the main point of our paper.
>
> With regards to Bassan and Katz, we agree that their work provides an interesting idea for provably correct local explanations (thus, in some cases, circumventing the need for an audit). By contrast, the results in our work are designed to apply to any local explanation scheme.
>
> W3. We agree that our work is limited to a certain type of "local explanation" and doesn't encapsulate explanation methods such as SHAP.  As we explained in our general rebuttal, we will make sure to carefully outline the scope of our paper in the final version. We will also emphasize in the final paper that studying other types of local explanation methods is an important and interesting direction for future work.

---

> > ### Comment · Reviewer_mYL9 · 2024-08-08
> >
> > I have read your responses and am satisfied with them. Overall, I believe the paper provides interesting contributions and could be a good fit for NeurIPS 2024. Adding a discussion on related work and clarifying the supported type of explanations will further improve the paper.

---

### Official Review · Reviewer_6haQ · 2024-07-12

**Soundness:** 2
**Presentation:** 2
**Contribution:** 2
**Rating:** 4
**Confidence:** 4

**Summary:**

This work studies an auditing framework in the eXplainable Artificial Intelligence (XAI) area. Specifically, the authors consider the scenario where a group of third-party auditors or users attempt to perform a sanity check on the provided explanations. The framework allows the auditors to query the model prediction and local explanations. Based on the proposed framework, this paper presents a theoretical analysis of the sample complexity of auditing.

**Strengths:**

This paper targets a very important aspect of XAI studies. It considers the deployment phase where users do not trust the provided explanations. This is a usually overlooked perspective in the XAI community.

**Weaknesses:**

1. This work focuses on local explanations defined in section 1.1 L48-49 and section 2.2 L155-162. These presumptions limit the scope of this paper to the surrogate model method (such as LIME, MUSE [1], etc.), where a glass-box explainer is used to approximate black box’ predictions in the neighborhood of input data. This greatly limits the impact of this work as such surrogate-model explanation methods only take a very small part of local explanation methods. Local explanations are not limited to surrogate model methods. It can refer to explanations regarding individual input samples instead of the entire data manifold or even regarding the model itself [2].
2. In the context of this paper, the authors claim that gradient-based explanations are surrogate model explanation methods (i.e. “local explanation method” under the definition of this paper) in L181-188. The authors define that $g_x(x) = (\nabla_xf(x))^Tx$, which is the summation of input x gradient attributions. This corresponds to the prediction $f(x)$ only if $f$ satisfies homogeneity [3]. On the contrary, suppose $\phi_f(x)\in\mathbb{R}^d$ is the explanation of SHAP, then $g_x(x):=(\phi_f(x))^T\mathbf{1} = f(x)$ can accurately reflects the prediction. Therefore, the definition of the explainers studied in this work is ambiguous and may require more rigorous considerations.

In summary of points 1 and 2, the formulation of the framework in this work is flawed.

3. There is no empirical verification of the proposed theoretical results, which significantly undermines the contribution of the theoretical analysis. Note that a continuous function is always bounded on the closed neighborhood of x. Therefore, it is essential to empirically test whether the proposed bounds are tight. A theoretical demonstration is also appreciated.

4. [minor] To stay consistent, “Lime” in L337 should be revised to “LIME”.

5. While the motivation that users/auditors may not trust the explanation system and want to audit the model is an interesting and realistic setup, the proposed framework lacks practical contributions. Specifically, the formalism described in L64-66 and L236-242. can be difficult to satisfy.

**Reference**

[1] Lakkaraju, H., & Bastani, O. (2020, February). " How do I fool you?" Manipulating User Trust via Misleading Black Box Explanations. In *Proceedings of the AAAI/ACM Conference on AI, Ethics, and Society* (pp. 79-85).

[2] Adebayo, J., Gilmer, J., Muelly, M., Goodfellow, I., Hardt, M., & Kim, B. (2018). Sanity checks for saliency maps. *Advances in neural information processing systems*, *31*.

[3] Hesse, R., Schaub-Meyer, S., & Roth, S. (2021). Fast axiomatic attribution for neural networks. *Advances in Neural Information Processing Systems*, *34*, 19513-19524.

**Questions:**

1. L51-52: Why do the authors claim LIME to be a gradient-based method?
2. Definitions 2.1 and 2.2 are limited to black box $f:\mathbb{R}^d\rightarrow \{\pm 1\}$. Can this constraint be relaxed to more general settings? For example, is it limited to a binary decision boundary?
3. The main theoretical results of Theorem 4.1 have many presumptions that are not justified. For example, why is the “user-specified” local error threshold assumed to be 1/3?

**Limitations:**

The limitations of this work are discussed in L194-197, where the authors admit that the definition “local explanation” is narrowed down to fit a. I agree with the limitation and appreciate the authors for clearly stating this issue. However, my concern is that this can be severe and greatly undermine the application scenarios of this work. More details are in the weakness section.

---

> ### Author Rebuttal · Authors · 2024-08-06
>
> We thank the reviewer for their review. We will begin by addressing the listed weaknesses
>
> 1. As we mention in our global rebuttal, we respectfully disagree that the considered explanations are too limited. LIME and Anchors are both reasonably utilized methods in practice, and more generally we believe our general formulation captures a meaningful subset of explanation methods.
>
> 2. We will clarify further what we meant in lines 181-188 in the final paper. We never state that $g_x(x) = (\nabla_x f(x))^t x$. We rather meant that $g_x$ should be some linear classifier that matches these coefficients. In particular, we would also fit a bias term so that $g_x$ agree with $f$ at the point $x$. Additionally, as mentioned in our global rebuttal, our framework does not consider SHAP.
>
> 3. We have left an empirical validation of our results for future work. We believe that the theory provided is sufficient for meaningfully arguing the difficulties posed by local explanations that are too local. We view the results of this paper as serving as a "theoretical demonstration."
>
> 4. Thank you, this will be fixed.
>
> 5. Could you please elaborate on what you found lacking in our framework? Additionally, what precisely about our framework can be "difficult to satisfy." Please see our global rebuttal for our general argument for the utility of our framework.
>
> Next, we will address the direct questions
>
> 1. We will correct this to be more precise in our general paper. While technically speaking, LIME does not use a gradient, we believe that at a high level it uses similar ideas to Gradient based methods by providing a local "linear-like" approximation to the general classifier.
>
> 2. We believe the binary classification setting is sufficiently relevant to be independently considered. However, we note that our lower bound directly extends to more general classification settings as binary classification is a subtask of multiclass classification. We also believe that our lower bound can be relatively easily adapted to regression by simply replacing the 0-1 loss used to define the local loss with an MSE based loss.
>
> 3. Could you specify specifically which assumptions are not justified? Furthermore, the user-specified threshold $\gamma$ is not set to $\frac{1}{3}$, it is instead assumed that $\gamma \leq \frac{1}{3}$. This is an extremely mild assumption considering that a local loss above $\frac{1}{3}$ would be considered egregiously inaccurate in most settings.

---

> > ### Comment · Reviewer_6haQ · 2024-08-08
> > **Thanks for the Responses**
> >
> > I appreciate the authors' responses and I clarify the questions as follows.
> >
> > W5: This originally refers to the settings of the auditing framework in real-world applications. For example, the requirement for numerous explanations from the auditee. But I agree this is a relatively minor point that does not involve technical issues.
> > Q3: This question is not regarding the conditions being too strong, but to ask why are they chosen as those values. This work focuses on realistic settings of problems. Therefore, discussions on the choices of parameters and their values can be beneficial and provide more insights.
> >
> > I have raised the scores accordingly.
> >
> > However, I disagree that empirical verifications should be left for future work. Given the realistic problem setups and motivations of this work, empirical validations of the theoretical results are very important. Besides, many of the claimed discoveries should not be difficult to verify. For example, "auditing an explainer requires an amount of data that is inversely proportional to its locality". For the completeness of the work, I still strongly encourage the authors to provide empirical verification as a part of the work, even for synthetic data. I will raise the score to 6 if this concern is resolved.

---

> ### Author Response · Authors · 2024-08-14
> **Response to reviewer**
>
> We thank the reviewer for reading and responding to our rebuttal. While we agree that empirical validation is an important direction, we don't agree that it is a straightforward verification task.
>
> Our main result is a lower bound that holds for any general explanation method (note that this necessarily includes manipulative explanations where the explainer might create explanations that appear as though they come from a gradient based method but in reality are maliciously modified) along with any general auditing method. Thus, experiments run for a specific pair of explanation and auditing methods do not serve as strong evidence for our theorem holding. Furthermore, the fact that local regions in high dimensional space can be extremely small implies that determining a ground truth value for the local loss itself could prove challenging.
>
> Due to these factors, we do not have enough time to include a meaningful empirical validation of our lower bound in this submission. Although we believe that our current results are sufficient for a complete paper, we respect the reviewers view that an empirical validation is a needed component.

---

### Official Review · Reviewer_cVCK · 2024-07-12

**Soundness:** 3
**Presentation:** 2
**Contribution:** 3
**Rating:** 6
**Confidence:** 4

**Summary:**

The paper proposes an auditing framework to verify the truthfulness of explanations by a third-party in scenarios where there is no trust. Bounds on sample complexity are provided that depend on the locality (minimum local mass) of the explanation. Further, the authors discuss that for gradient-based explanations in higher dimensions, locality tends to be small to achieve a reasonable explanation loss. Smaller locality increases the provided bounds on the amount of data required for the audit.

**Strengths:**

1. The topic of the paper is important for policymakers and the XAI research community in general, as it suggests that in natural scenarios where there is no trust, it is difficult to verify whether the local explanation is truthful to the model without actually knowing the model.
2. The paper provides upper and lower bounds on sample complexity for auditing local explanations.
3. The analysis includes gradient-based explanations and discusses how to generalize to other methods, including LIME and Anchors.

**Weaknesses:**

1. Regarding the soundness of the auditing framework, could you please comment on the motivation for the company to provide all of the required data (especially local regions) to the auditor? When the requested dataset is sufficiently large, the third-party company could potentially recover the model along with all the classifier outputs, local regions, and other information.
2. Figure 1 is hard to fully understand without reading the paper. It’s not intuitive which data points are explained and why, in panel (b), there is sufficient data for the audit. Could you please provide more explanation in the figure caption or simplify the figure?
3. Section 5 and Theorem 5.1 present an existence proof. However, the example considered in Figure 2 (a) is very specific. Can you elaborate on how often you expect this data distribution to occur in real-world datasets or discuss the value/range of locality for other likely data distributions?

Minor:
1. $E(f, x_i)$ is used in Section 1 but defined in Section 2.
2. Please specify $\epsilon_1$, $\epsilon_2$, $\delta$, and $\gamma$ in Theorem 4.2 or mention that you are operating under the same conditions as in Theorem 4.1, if applicable. Also, Algorithm 1 is referenced before it is defined.

**Questions:**

1. Do you have any guidance or suggestions on how users should choose the local error threshold, $\gamma$?
2. How useful are the bounds in Section 4 for reasonable values of locality and low explanation loss, such as when the decision boundary is nearly locally linear? In this case, can you provide any empirical estimates on the lower and upper bounds in Theorems 4.1 and 4.2?

**Limitations:**

The authors discuss the limitations of the paper with respect to the explanation algorithms.

---

> ### Author Rebuttal · Authors · 2024-08-06
>
> We thank the reviewer for their detailed review. We will first respond to the listed weaknesses.
>
> 1. As we mention in our global rebuttal, we believe our setting reflects cases where an Auditor might have access to a set of explained cases by default. For example, applicants to a bank for loans could, in principle, pool their explanations to provide a dataset with which an Auditor could operate within our setting. We will discuss further in the final version of our paper, and will include examples of cases where our setting could plausibly apply.
>
> 2. We appreciate the feedback on figure 1, we will certainly add more detail to the caption and simplify the figure by removing the green decision boundary and including enough points to visually demonstrate the concept of "enough data for auditing."
>
> 3. The key issue posed by Figure 2 is the curvature in the data manifold. Thus, we believe that cases with high dimensional and highly curved data manifolds would pose similar issues as the decision boundary involved would seldom be linear. A detailed empirical investigation of this is left as a direction for future work.
>
> 4-5. Thank you, these will both be fixed.
>
> Regarding the direct questions:
>
> 1. We believe $\gamma$ to be largely application based, and for this reason chose to leave it as a user-specified parameter. However, one basic guiding principle might be to set $\gamma$ similarly to the general loss of the classifier being explained. For example, it doesn't seem necessary to require local explanations with a corresponding local loss of 0.001% for a classifier with a 30% misclassification rate.
>
> 2. Our local and upper bounds are designed for arbitrary, potentially complex classifiers. We chose this as the default starting point for theoretical analysis. We believe that studying more restricted cases of classifiers could be very interesting and promising. For example, we believe that ensuring a classifier satisfies some degree of smoothness (i.e. close to locally linear decision boundaries) could plausibly greatly reduce the amount of data needed for auditing.

---

> > ### Comment · Reviewer_cVCK · 2024-08-12
> >
> > Thank you to the authors for the response. I have read the authors’ reply and the other reviews, and I will keep my score.

---

### Official Review · Reviewer_KRVz · 2024-07-12

**Soundness:** 3
**Presentation:** 3
**Contribution:** 3
**Rating:** 7
**Confidence:** 3

**Summary:**

This paper provides theoretical results on how many queries are required for an auditing framework for local explanations of machine learning algorithms (e.g., neural networks).

**Strengths:**

The paper is well motivated with a widely interesting and relevant topic. The approach is theoretical, and rigor is provided through proofs in appendices. The authors connect their work to popular algorithms: Gradient based approaches, Lime, and Anchors.

**Weaknesses:**

The authors acknowledge that the local loss estimation is, in their argument, only a necessary but not sufficient condition for trust. They do not establish or reference any evidence that manipulation would result in the local loss as a good indicator of untrustworthiness. As a result, the analysis serves more as a potential validation scheme for the limited types of algorithms that meet their linearity requirement (e.g. Anchors or LIME). This drastically narrows the scope and implications of their analysis. Unless this can be firmly established the title, abstract, and conclusions of the paper should be amended to reflect the correct scope of its claims
Furthermore, there are plenty of reasons (and examples) where interpretation methods are demonstrated to be frail to the *input* (e.g. Ghorbani). This would likely not pass the audit but would not be evidence that the $E$ has been altered. I think this speaks to some confusion in the setup of the paper as to what “trust-worthiness” is. The authors present it as trust between the user and provider rather than trust in the robustness of the explainability metric which is, I argue, closer to what their results seem to reflect.
Additionally, it is not clear to me that this is the only way to test explainability metrics with limited access (e.g. data points, classifier outputs, and local explanations).  For example, these metrics are popular because they match so well to human “expert” knowledge. You show someone the pcitre of the shark from Smilkov et al and they agree that they see “shark-like” shapes in the SmoothGrad output. Consequently, one could imagine the case where sampling in a *non-local* fashion would trace whether the same features matching human “experts” appear.
Ultimately the proposed methodology seems entirely impractical (which is sort of the point).

**Questions:**

-	Can you establish or provide a reference hat show how the local loss / explainability loss is a good indicator for manipulation from an adversarial attack (or disingenuity design)? This is critical for the scope and implications of your analysis.
-	Please comment on whether the proposed auditing scheme is indeed the only way to establish a local explainer (fitting your constraints) taking into account the suggestions above.
-	Can you comment on how the results might
-	L 117 please change reference to Shap to the acronym SHAP
-	L 337 please change reference to the Lime algorithm to its acronym: LIME
-	L 440 where does the 2592 come from in the denominator for the bound on $n$?
-	How well would the framework work if you chose some K examples to audit around instead of drawing sample i.i.d.?

**Limitations:**

-	Analysis is for a binary classifier

---

> ### Author Rebuttal · Authors · 2024-08-06
>
> We appreciate the review and the detailed questions. In order
>
> 1. As we mention in our global rebuttal, we do not believe the local loss is sufficient for preventing adversarial attacks. We rather believe that maintaining a low local loss is one of many necessary components required for a trustworthy explanation. We intentionally do not define trustworthiness, we are rather just showing that auditing one part of it can be difficult from a data perspective.
>
> 2. Our scheme isn't intended to encapsulate ways to "establish a local explainer." It is rather meant to reflect plausible settings where an Auditor might seek to audit explanations provided by some entity that desires to protect information about their models.
>
> 3. Could you finish this question in a comment? We'd be happy to respond during the next period of the review.
>
> 4-5. Thank you, we will fix these.
>
> 6. This is a constant chosen simply for mathematical reasons. It is merely designed to make the algebra within our bounds to work out and is not tight (as our goal is to simply provide an asymptotical lower bound). Replacing it with a different constant would simply result in changing values of $\epsilon$ and $\delta$. Note that for small values of $\lambda$ (in high dimensional data), this constant would be easily dominated by $\lambda$.
>
> 7. This is an interesting question. This would depend a lot on the manner in which the K examples are chosen. While it appears that querying samples from a single local region would allow for very quick estimation of the local loss, we believe that ensuring that these examples are "in-distribution" would pose a technical challenge. In particular, the auditor would have to have some notion of what constitutes a "realistic" example. We believe this is an interesting direction for future work.
>
> We would additionally like to respond to some of the other points raised in the "Weaknesses" section. As we mention in our global rebuttal, we don't feel that our setting is too limited to bear relevance, and we also aren't quite sure what the "linear assumption" being referenced is. As we mentioned in answer 1 and in our global rebuttal, we do not define trust as the local loss. We rather argue that a low local loss is necessary but not sufficient for there to be trust.
>
> Finally, as we mentioned in answer 2, our paper is designed to examine a natural default way in which a set of explanations might be audited. In fact, one of the implications of our results is precisely the necessity for other methodologies.

---

> > ### Comment · Reviewer_KRVz · 2024-08-08
> >
> > I appreciate and have read the response. My apologies for any confusion regarding the unfinished question.

---

### Author Rebuttal · Authors · 2024-08-06

We thank the reviewers for their detailed and thoughtful reviews. It appears that there are 3 main points of contention regarding our paper. First, that the set of "local explanations" being considered is either too limited or not carefully enough analyzed, second, that the local loss is not a good indicator of a model's trustworthiness, and third, that the mode of limited interaction between the auditor and the explanation provider is not sufficiently realistic. We will address these three points separately.

We define a local explanation broadly as a local classifier coupled with a local region. We will further clarify in our paper that this does NOT include "local" methods such as SHAP (in which no local region is provided). However, this definition does include LIME, Anchors, and certain kinds of gradient based methods (where a gradient implies that a linear classifier is implicitly being used). We believe that these listed methods are sufficiently used by practitioners to be worth consideration. While our work only considers linear and constant local classifiers, we believe this to be a strength rather than weakness as our work is centered on providing a lower bound for auditing (which means our bounds will carry over to more complex classes of local classifiers as well).

With regards to the local loss, we will further clarify (as we did in line 55 of our paper) that maintaining a low local loss is a necessary but not sufficient condition for being a trustworthy explanation. We completely agree that a low local loss does not imply that an explanation is trustworthy. We are rather claiming that it is self evident that explanations with a large local loss are intrinsically meaningless and untrustworthy. Because of this, our auditing framework encapsulates one necessary component that any auditing framework most address, and this means our lower bound has implications beyond our setting. In particular, circumventing the difficulties posed by our lower bound must require one of our assumptions to be broken which would require either further access for the auditor, or a different kind of explanation being provided.

Finally, our mode of interaction between the explainer and the auditor is inspired by cases of collective action. In principle, we believe that an auditor could obtain a sample of explanations by aggregating outputs provided across a large sample of users. For example, in the case where a bank provides local explanations for loan approvals to applicants, an auditor could (with proper consent) aggregate the explanations provided for a sample of applicants and audit the bank. Note that this would not require any further information from the bank beyond the explanations it is already required to provide.

---

### Decision · Program_Chairs · 2024-09-25

**Decision:**

Accept (poster)

**Comment:**

The paper provides valuable insights into the challenges of auditing local explanations and establishes a foundation for future work in this area, and I thus recommend accepting it.
The authors' rebuttal was appreciated by the reviewers. They acknowledged the need for empirical validation but argued that such validation is challenging given the generality of their theoretical results -- please make sure to discuss this aspect in your final version, should the paper make the cut.  Please also make sure to improve the discussion of related work and to clarify the scope of your study.